# UltraEdit: Training-, Subject-, and Memory-Free Lifelong Editing in Language Models

**Xiaojie Gu**                                                              *peettherapynoys@gmail.com*
*Independent Researcher*

**Ziying Huang**                                                              *m74775466@gmail.com*
*Independent Researcher*

**Jia-Chen Gu**                                                              *gujc@ucla.edu*
*UCLA*

**Kai Zhang**[*]                                                              *drogozhang@gmail.com*
*The Ohio State University*

**Reviewed on OpenReview:** *https://openreview.net/forum?id=GoJLp3BlRV*

## Abstract

Lifelong learning enables large language models (LLMs) to adapt to evolving information by continually updating their internal knowledge. An ideal system should support efficient, wide-ranging updates while preserving existing capabilities and ensuring reliable deployment. Model editing stands out as a promising solution for this goal, offering a focused and efficient way to revise a model's internal knowledge. Although recent paradigms have made notable progress, they often struggle to meet the demands of practical lifelong adaptation at scale. To bridge this gap, we propose ULTRAEDIT, a *training-*, *subject-*, and *memory-free* approach that is well-suited for ultra-scalable, real-world lifelong model editing. ULTRAEDIT fundamentally differs from traditional paradigms by computing parameter shifts in one step using only a hidden state and its gradient, making the approach simple yet efficient. To improve scalability in lifelong settings, ULTRAEDIT employs a *lifelong normalization* strategy that continuously updates feature statistics across turns, allowing it to adapt to distributional shifts and maintain consistency over time. ULTRAEDIT achieves editing speeds more than **7×** **faster** than the previous state-of-the-art method, while requiring **4× less VRAM**. This makes it the **only** method currently capable of editing a 7B LLM on a 24GB consumer-grade GPU. Furthermore, we construct ULTRAEDITBENCH, the largest dataset in the field to date with over **2M** editing pairs, and demonstrate that our method supports up to **2M** edits while maintaining high accuracy. Comprehensive experiments on five datasets and six models show that ULTRAEDIT consistently achieves superior performance across diverse model editing scenarios, taking a further step towards safe and scalable lifelong learning. Our code is available at: `https://github.com/XiaojieGu/UltraEdit`

## 1 Introduction

Lifelong learning (also known as continual learning (Shi et al., 2024; Zheng et al., 2025b)) is essential for enabling large language models (LLMs) to continuously adapt to evolving knowledge and real-world dynamics. Despite its importance, scalable and reliable lifelong adaptation remains challenging in practice (Wu et al., 2024a). Retraining is prohibitively expensive and slow, making it unsuitable for frequent updates (Chen et al., 2023). Meanwhile, existing lifelong learning approaches often suffer from catastrophic forgetting, or depend

---

[*]Corresponding author.

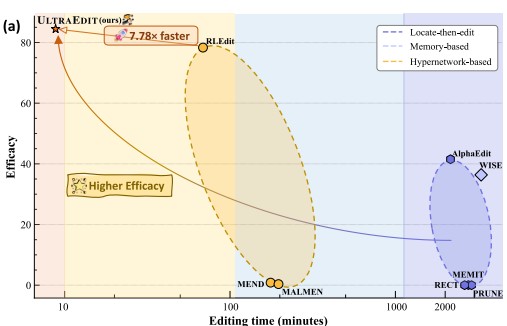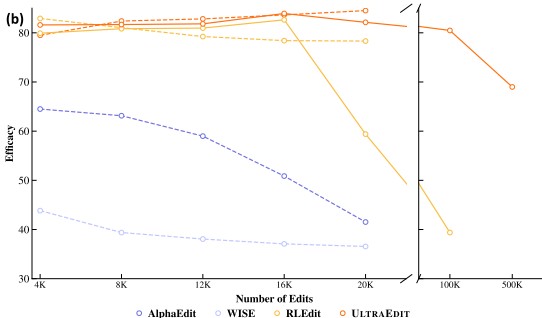

Figure 1: (a) Average Efficacy and editing time of different solutions on 20K edits from ZsRE, evaluated across GPT-J, Mistral, and LLaMA-3. (b) Variation in average Efficacy as edits accumulate. Dashed lines represent performance on the ZsRE dataset across GPT-J, Mistral, and LLaMA-3, while solid lines represent results on the WikiBigEdit dataset with LLaMA-3.

on retrieval-augmented generation (RAG; Jimenez Gutierrez et al. (2024); Gutiérrez et al. (2025)), which may potentially introduce conflicts between retrieved content and the model's internal knowledge (Xie et al., 2023). These limitations suggest that current paradigms may not fully meet the demands of lifelong deployment, highlighting the need for more targeted and efficient mechanisms for continual knowledge integration (Aljundi et al., 2019).

One promising solution is model editing (De Cao et al., 2021; Meng et al., 2023; Yao et al., 2023), which enables targeted modifications to a model's internal knowledge while leaving unrelated information unaffected, making it especially suited for continual updates over time (Hartvigsen et al., 2023; Hu et al., 2024). Some methods enhance this process by training auxiliary networks (Mitchell et al., 2022a; Li et al., 2025) to guide how model parameters are adjusted in response to new information. Others rely on strong structural assumptions, such as *subject-centric* representations (Meng et al., 2023; Li et al., 2024) or carefully formatted input prompts (Zheng et al., 2023; Zhong et al., 2023b), which tie them to handcrafted data pipelines (Deng et al., 2025; Zhong et al., 2025) and limit both generalization and practical feasibility. In addition, many approaches depend on external memory (Wang et al., 2024b) to store edits separately from model parameters; while this localizes changes and avoids direct parameter overwriting, it still requires training to update memory entries and introduces substantial overhead that grows with the number of edits, thereby reducing scalability in large-scale or high-frequency editing scenarios (Bi et al., 2024). To address the issues mentioned above, we propose ULTRAEDIT: a simple yet efficient *training-*, *subject-*, and *memory-free* method.

ULTRAEDIT introduces a *lifelong normalization* mechanism that continuously updates feature statistics during editing. By continually adjusting the mean and variance of concatenated hidden states and gradients, the method implicitly enforces a stable feature geometry that preconditions the least-squares system used for parameter updates. This normalization acts as a form of online whitening: it equalizes feature scales, prevents representation drift from amplifying update magnitudes, and reduces the risk of new edits overwriting previously acquired knowledge. The entire procedure relies solely on simple linear algebra over editing features and requires no iterative optimization or external memory structures. As a result, ULTRAEDIT supports efficient and large-scale edits with strong stability and consistency, making it practical for real-world deployment (Zheng et al., 2025a; Grimes et al., 2025). Existing paradigms, by contrast, are prone to the *Edit Collapse* phenomenon (Yang et al., 2024b; Wang et al., 2025), which refers to a sharp decline in editing stability and effectiveness as the number of edits or editing turns grows. As shown in Figure 1, ULTRAEDIT not only avoids this collapse but also achieves significantly faster editing speeds while maintaining consistent performance in ultra-large-scale lifelong editing scenarios. A comparison between ULTRAEDIT and other solutions is provided in Table 1.

To evaluate our method at its full potential and push the boundaries of large-scale model editing, we introduce our benchmark ULTRAEDITBENCH, a factual QA benchmark constructed from Wikidata triples, comprising over 2 million complete editing pairs. In addition to this new benchmark, we evaluate the effectiveness and scalability of ULTRAEDIT on four widely used model editing datasets, namely zsRE (Levy et al., 2017), FEVER (Thorne et al., 2018), WikiBigEdit (Thede et al., 2025), and UnKE (Deng et al., 2025), across six diverse models including GPT (Wang & Komatsuzaki, 2021), Mistral (Jiang et al., 2023), LLaMA (Grattafiori

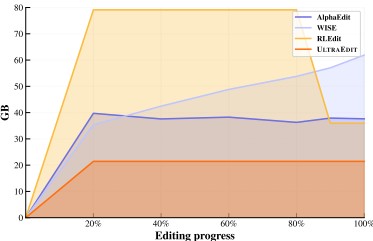

Figure 2: VRAM usage over the course of 20K edits on the ZsRE dataset using different methods with Mistral-7B.

Table 1: Comparison between the UltraEdit and three prevailing model editing solutions. ✓indicates "yes" or well-supported, while ✗denotes "no" or badly-supported.

| Solution | Training-free | Subject-free | Memory-free | Lifelong scalability |
|---|---|---|---|---|
| Locate-then-edit | ✓ | ✗ | ✓ | ✗ |
| Memory-based | ✗ | ✗ | ✗ | ✗ |
| Hypernetwork-based | ✗ | ✓ | ✓ | ✗ |
| UltraEdit | ✓ | ✓ | ✓ | ✓ |

et al., 2024), Qwen (Yang et al., 2024a), Phi (Abdin et al., 2024), and Gemma (Team et al., 2025). Our results show that UltraEdit achieves new state-of-the-art performance across most editing scenarios. In addition to its strong empirical performance, UltraEdit demonstrates remarkable efficiency: it achieves over 7× faster editing speeds and uses less than one-fourth of the GPU memory compared to prior leading baselines, as show in Figure 1 (a) and Figure 2. Notably, it is the only method to date capable of reliably editing a 7B-scale model on a standard 24GB consumer GPU, making it uniquely practical for real-world deployment. Moreover, UltraEdit demonstrates the ability to scale to 2 million edits while preserving model stability, underscoring its promise for ultra-large-scale lifelong model editing. Our contributions are four-fold:

- We identify and analyze the shortcomings of three dominant model editing paradigms under large-scale lifelong settings, providing insights for developing more advanced editing methods.
- We propose UltraEdit, a novel training-, subject-, and memory-free editing solution that performs stable updates through lifelong normalization. UltraEdit achieves over **7× faster** editing and **4× less VRAM** compared to previous state-of-the-art methods.
- We construct UltraEditBench, currently the largest dataset for model editing, comprising more than **2 Million** editing pairs to facilitate research on lifelong ultra-large-scale model editing.
- Comprehensive experiments on five benchmarks and six models, demonstrating UltraEdit's superior performance and scalability to **2 Million edits**.

## 2 Related Work

### 2.1 Lifelong Learning

Lifelong learning, also known as continual learning, aims to develop models that can continuously learn from a stream of tasks while preserving knowledge acquired in earlier stages (De Lange et al., 2021; Shi et al., 2024). The primary challenge lies in overcoming catastrophic forgetting, where new learning interferes with previously acquired knowledge. To address this challenge, existing approaches generally fall into three categories: regularization-based methods (Kirkpatrick et al., 2017; Zenke et al., 2017), which focus on preserving important parameters; replay-based methods (Rebuffi et al., 2017; Shin et al., 2017), which revisit past information during training; and architecture-based methods (Rusu et al., 2016; Mallya & Lazebnik, 2018), which dynamically adjust the model structure.

### 2.2 Model Editing Paradigm

**Hypernetwork-based** methods (Mitchell et al., 2022a; Tan et al., 2024; Li et al., 2025; Gu et al., 2026) treat model editing as a meta-learning problem by training a separate neural network to predict parameter shifts. This auxiliary network operates independently from the base model and learns to project editing inputs into effective weight updates. Once trained on a collection of edits, it enables quick application of new updates without re-optimizing for each case. However, the hypernetwork remains fixed while the underlying model continues to evolve as edits accumulate. This growing mismatch can lead to degraded editing performance over time. Furthermore, the need for additional training data limits the practicality of these methods in scenarios requiring rapid or continual updates.

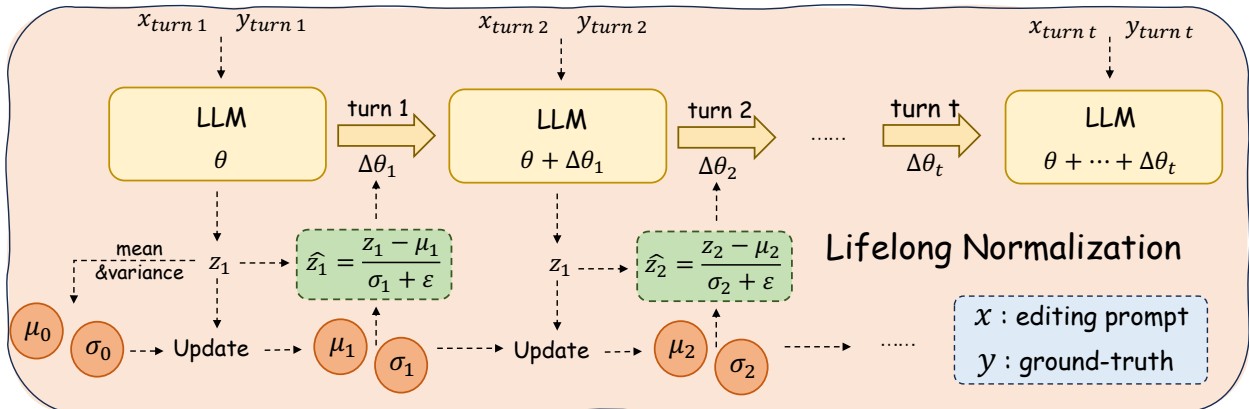

Figure 3: This figure illustrates the lifelong editing workflow of ULTRAEDIT, where parameter shifts are applied iteratively across turns using a lifelong normalization mechanism that maintains running statistics of editing-instance features to ensure stable and consistent model behavior over time.

**Locate-then-edit** methods (Dai et al., 2022; Meng et al., 2023; Wang et al., 2024a; Gupta et al., 2024; Fang et al., 2025; Pan et al., 2025) rely on the presence of an explicit subject or entity in the input to identify which internal components of the model are primarily responsible for storing the corresponding piece of knowledge. They then apply targeted perturbations, typically through computationally intensive iterative optimization, to enforce the desired output while minimizing side effects on unrelated behaviors and representations. Although effective for isolated edits, these methods often become unstable in lifelong settings, as repeated updates to overlapping parameters can accumulate and lead to interference or degradation of previously edited knowledge.

**Memory-based** approaches (Dong et al., 2022; Mitchell et al., 2022b; Zheng et al., 2023; Hartvigsen et al., 2023; Yu et al., 2024; Wang et al., 2024b) enhance the model with an external memory that stores edits separately from the core parameters. New knowledge is incorporated by adding or modifying memory entries, which localizes changes and minimizes side effects on the base model. However, these methods still require training to update memory representations or routing, and inference must be rerouted through the memory. In addition, they maintain one entry per edit, causing memory consumption to grow linearly with the number of edits, which limits their practicality in lifelong model editing with high-frequency or large-scale updates.

## 3 Methodology

### 3.1 Preliminary

We consider a pre-trained language model $f_\theta : \mathcal{X} \to \mathcal{Y}$ with parameters $\theta$. *Model editing* is the task of modifying this model's parameters to change its behavior on a specific input while leaving the rest of its behavior intact. Formally, an editing instance is a pair $(x_e, y_e) \in \mathcal{X} \times \mathcal{Y}$, where $x_e$ is an edit query and $y_e$ is the ground-truth output for that input. The goal is to find a parameter set $\theta'$ such that the edited model $f_{\theta'}$ produces the desired output $y_e$ for the edit query $x_e$ (*Efficacy*). In addition, the model should respond appropriately to a set of equivalent queries $\mathcal{E}(x_e) \subset \mathcal{X}$, which includes semantically similar variations of the original query (*Generalization*), while ensuring that its behavior on a set of unrelated queries $\mathcal{U}(x_e) \subset \mathcal{X}$, which are not associated with the edited knowledge, remains unaffected (*Specificity*).

In a lifelong model editing scenario, the model is updated iteratively across a sequence of editing turns. At each turn $t$, $n$ editing instances $\{(x_e^{(t,i)}, y_e^{(t,i)})\}_{i=1}^n$ is applied to the current model $f_{\theta^{(t-1)}}$, resulting in a new model $f_{\theta^{(t)}}$. Each turn of edits is performed on the model that has already incorporated all previous edits, making stability across turns critically important.

---

**Algorithm 1:** ULTRAEDIT

---

**Input:** Model $f_\theta$, editing instances $\{(x_i, y_i)\}_{i=1}^n$, editable modules $\mathcal{M}$
**Output:** Edited model $f_{\theta'} = f_\theta + \{\Delta\theta_m\}_{m \in \mathcal{M}}$
**foreach** *instance* $(x_i, y_i)$ **do**

    Run forward pass to extract hidden state $h_i$ at each module $m \in \mathcal{M}$;
    Run backward pass to compute gradient $\nabla y_i$ at each module;
    Concatenate editing feature: $z_i \leftarrow [h_i \,\|\, \nabla y_i]$;
    Update running mean $\mu$, variance $\sigma$ using lifelong rules;
    Cache $z_i$ for each module;

**foreach** *module* $m \in \mathcal{M}$ **do**

    Load cached feature vectors $\{z_i\}_{i=1}^n$ for module $m$;
    Normalize: $\hat{z}_i \leftarrow \frac{z_i - \mu}{\sigma + \varepsilon}$;
    Split: $[\tilde{h}_i \,\|\, \tilde{v}_i] \leftarrow \hat{z}_i$;
    Compute scaled update: $v_i = -\eta \cdot \|\tilde{h}_i\|^2 \cdot \tilde{v}_i$;
    Stack $H \leftarrow [h_1^\top, ..., h_n^\top]^\top$, $V \leftarrow [v_1^\top, ..., v_n^\top]^\top$;
    Compute optimal closed-form solution: $\Delta\theta_m \leftarrow (H^\top H + I)^{-1} H^\top V$;
    Apply: $\theta'_m \leftarrow \theta_m + \Delta\theta_m$;

**return** *Post-edited model* $f_{\theta'}$

---

## 3.2 UltraEdit

**Motivation** A central challenge in lifelong editing is to update model parameters in a way that is both *targeted* and **stable**. Prior paradigms highlight this trade-off in different ways: hypernetwork-based methods generate parameter updates via an auxiliary network, but they require costly pretraining and often generalize poorly; locate-then-edit methods directly modify causal mediators, typically identified using the subject entity in the editing instance, which provides precision but relies on per-case localization and iterative optimization; and memory-based approaches preserve past edits for reuse, yet their external memory structures inevitably expand and become increasingly expensive to maintain. These limitations make it difficult to support scalable, high-frequency updates in real-world lifelong learning scenarios. ULTRAEDIT addresses this challenge by exploiting intrinsic signals already present in each editing instance and combining them with closed-form optimization and lifelong normalization. This lightweight and training-free design avoids external memory or iterative procedures, while ensuring edits can be applied efficiently and consistently across long trajectories.

**Principle** At each editing turn $t$, ULTRAEDIT processes a batch of $n$ editing instances by extracting two complementary signals from a designated editable module (e.g., a transformer feedforward layer). A *forward hook* records the hidden state $h_i \in \mathbb{R}^d$ at the token position corresponding to the ground-truth answer (the unmasked label position), which specifies **where** the target knowledge is represented in the model. This hidden state anchors the edit to the correct semantic subspace and provides a coordinate system for locating the desired modification, and does not introduce information leakage, as the label is explicitly part of the editing target (Li et al., 2025; Fang et al., 2025). In parallel, a *backward hook* obtains the gradient vector $\nabla y_i \in \mathbb{R}^{d'}$ with respect to the same ground-truth output, derived from the supervised loss. This gradient indicates **how** the model's parameters should move to internalize the desired knowledge, thus encoding the direction of change required by the editing constraint. By concatenating these two signals, we construct a unified editing feature $z_i = [h_i \,\|\, \nabla y_i] \in \mathbb{R}^{d+d'}$ that simultaneously encodes the **location** (via hidden state) and **direction** (via gradient) of the update. This combination is crucial: the hidden state alone cannot determine how the model should change, and gradients alone lack information about the representational subspace in which the change should occur. Their union therefore provides a complete minimal sufficient statistic for closed-form estimation and enables the method to generate parameter shifts without auxiliary training or iterative optimization. This feature representation also forms the basis for the lifelong normalization mechanism described below.

**Design** Building on the joint feature $z_i = [h_i \| \nabla y_i]$, we introduce a *lifelong normalization* mechanism that co-normalizes hidden states and gradients across editing turns. In lifelong editing, the distributions of hidden activations and gradients inevitably drift as each edit alters the model's internal representations (Tan et al., 2024). Without correction, this drift produces inconsistent feature scales, ill-conditioned least-squares systems, and unstable or interfering updates over long trajectories. To address this challenge, we maintain running statistics $(\mu, \sigma)$ over all past editing features and normalize each new instance as follows:

$$\hat{z}_i = \text{Norm}(z_i) = \frac{z_i - \mu}{\sigma + \varepsilon}, \tag{1}$$

where $\mu$ and $\sigma$ are the running mean and standard deviation, and $\varepsilon$ is a small constant for numerical stability. This normalization plays a role analogous to whitening or preconditioning: it stabilizes the effective learning rate, ensures comparability of features across turns, and preserves a well-conditioned geometry for subsequent parameter estimation. Crucially, because hidden states and gradients are normalized jointly, their relative scaling remains stable, preventing one signal from dominating the other as the model evolves.

At each editing turn $t$, let $\{z_i\}_{i=1}^n$ denote the concatenated feature vectors from $n$ editing instances. The turn-wise mean and variance are computed as $\bar{z} = \frac{1}{n}\sum_{i=1}^n z_i$ and $\text{Var}(z) = \frac{1}{n}\sum_{i=1}^n (z_i - \bar{z})^2$, respectively. We then compute the difference $\delta = \bar{z} - \mu$, accumulate squared deviations with $s^2 \leftarrow s^2 + n \cdot \text{Var}(z) + \frac{Nn}{N+n} \cdot \delta^2$, and update the sample count as $N \leftarrow N + n$.

The running mean and standard deviation are updated via:

$$\mu \leftarrow \mu + \frac{n}{N+n} \cdot \delta, \tag{2}$$

$$\sigma \leftarrow \sqrt{\frac{s^2}{N+n-1+\varepsilon}}. \tag{3}$$

At the first turn, since no prior statistics exist, we initialize the running mean and standard deviation as $\mu_0 = \bar{z}$ and $\sigma_0 = \sqrt{\text{Var}(z) + \varepsilon}$.

After normalization, the vector $\hat{z}_i \in \mathbb{R}^{d+d'}$ is split into two components:

$$[\tilde{h}_i \| \tilde{v}_i] = \hat{z}_i, \tag{4}$$

where $\tilde{h}_i \in \mathbb{R}^d$ is the normalized hidden state, and $\tilde{v}_i \in \mathbb{R}^{d'}$ the normalized gradient value.

To adaptively scale the influence of each editing sample, following (Mitchell et al., 2022a; Tan et al., 2024; Li et al., 2025), we employ a scaling mechanism based on the magnitude of its normalized hidden state. Specifically, for each sample $i$, the scaled update direction is computed as:

$$v_i = -\eta \cdot \|\tilde{h}_i\|^2 \cdot \tilde{v}_i, \tag{5}$$

where $\eta$ is a global scaling factor analogous to a learning rate. The scaling reflects the saliency of the hidden representation and determines the strength of the corresponding edit. The multiplicative factor $\|\tilde{h}_i\|^2$ adaptively scales the update according to the saliency of the hidden representation, ensuring that edits are strongest where the model already encodes the target concept. This mechanism further reduces interference by down-weighting noisy or weakly relevant gradients.

Let $H \in \mathbb{R}^{n \times d}$ be the matrix of unnormalized hidden state and $V \in \mathbb{R}^{n \times d'}$ the matrix of scaled update vectors $v_i$. The final parameter shift is obtained by solving a regularized least-squares problem that minimizes both the reconstruction error and the update norm:

$$\min_{\Delta\theta} \|H\Delta\theta - V\|^2 + \|\Delta\theta\|^2, \tag{6}$$

where $\theta' \in \mathbb{R}^{d \times d'}$ is the target weight perturbation. The optimal closed-form solution is given by:

$$\Delta\theta = (H^\top H + I)^{-1} H^\top V. \tag{7}$$

This solution yields the minimum-norm update that best satisfies all editing constraints simultaneously. Geometrically, the projection induced by $(H^\top H + I)^{-1} H^\top$ aligns updates with the subspace spanned by hidden activations, suppressing components that would conflict with earlier edits and thereby mitigating interference. Combined with lifelong normalization, which keeps this geometry well-conditioned across turns, this projection-based update mechanism ensures that edits remain targeted, stable, and compatible throughout long editing sequences. The resulting edited model parameters are obtained by directly applying the shift, i.e., $\theta' = \theta + \Delta\theta$, thereby completing the model editing process.

**Theoretical Justification**    We provide a theoretical justification for why our *Lifelong Normalization* serves as a sufficient mathematical proxy for the explicit covariance preservation ($C_0$) used in Locate-then-Edit methods like MEMIT and AlphaEdit. MEMIT formulates model editing as a constrained least-squares problem, relying on the uncentered covariance matrix $C_0$ of the pre-trained keys, denoted as $K$, to minimize interference. (Here, $K$ represents the input keys to the layer in MEMIT's formulation, which corresponds directly to the hidden states $H$ in our notation.) Geometrically, $C_0$ acts as a metric tensor in the update formula $\Delta \propto (C_0 + KK^\top)^{-1}$, effectively defining the Mahalanobis distance within the parameter space. This mechanism penalizes deviations along the principal components of $C_0$, which correspond to directions of high variance where the model possesses rich knowledge, thereby forcing the update vector $\Delta$ into the null space or low-variance directions of the old knowledge.

Lifelong Normalization addresses the computational bottleneck of explicit covariance calculation by dynamically maintaining first- and second-order statistics to perform a standardization transformation. Specifically, we implement a joint normalization on the combined vector of hidden states and error gradients, denoted as $z = [h \parallel \nabla y]$. This acts as an *online preconditioning* mechanism that stabilizes the spectral properties of the feature covariance matrix. By centering and scaling this joint distribution, the process effectively whitens the feature space $H$, ensuring that the condition number $\kappa(H^\top H + I)$ remains bounded despite the distributional drift inherent in lifelong learning. Originally based on limited samples, the running statistics $(\mu, \sigma)$ progressively converge to a robust representation of the global feature distribution. The key geometric significance of this distributional transformation is that it implicitly reshapes the covariance matrix of old knowledge $\hat{C}_0$ into an approximate identity matrix, $\hat{C}_0 \approx I$. This whitening transformation establishes a direct mathematical equivalence, causing the complex Generalized Least Squares (GLS) problem in MEMIT to degenerate into a computationally efficient Ordinary Least Squares (OLS) problem within the normalized space. Substituting the whitened unit covariance $\hat{C}_0 \approx I$ into MEMIT's analytical solution logic naturally simplifies the regularization term $(C_0 + KK^\top)^{-1}$, where $K$ is now replaced by the normalized hidden states $H$, into the form $(I + HH^\top)^{-1}$ adopted by ULTRAEDIT (Eq. 7). Theoretical analysis reveals that this mature calibration prevents anisotropy in the effective Hessian, guaranteeing that parameter updates approximate an *orthogonal projection* onto the active subspace rather than projecting non-orthogonally into the null space of prior knowledge. Therefore, ULTRAEDIT minimizes the spectral norm of perturbations on unrelated subspaces, effectively serving as a surrogate for the weighted orthogonality of MEMIT and ensuring resistance to cumulative interference as statistical estimation matures.

ULTRAEDIT's complete pipeline is illustrated in Figure 3, and the pseudocode of the ULTRAEDIT editing procedure in one turn is provided in Algorithm 1. The practical applicability of ULTRAEDIT is discussed in the real-world lifelong application statement provided in the Appendix A.

## 4    Experiments

### 4.1    UltraEditBench-2M Construction

We construct ULTRAEDITBENCH using entity–relation–object triples from the Wikidata5M (Wang et al., 2021) knowledge base. For each triple, we treat the object as the ground-truth answer and use the GPT-4o-mini model in a zero-shot setting to generate corresponding factual questions based on the subject and relation. To promote linguistic diversity, we apply constraints on question length and phrasing during generation. To ensure data quality, we perform random spot checks on a subset of samples to verify factual accuracy, linguistic fluency, and alignment between questions and answers.

Table 2: Results on three datasets across three models and bold values indicate the best results. *Eff.* denotes *Efficacy*, *Gen.* denotes *Generalization*, and *Spe.* denotes *Specificity*. Δ indicates the performance difference between ULTRAEDIT and the previous best method. ULTRAEDIT denotes results under the same number of edits as other baseline methods, while ULTRAEDIT* represents performance in an ultra-large-scale editing scenario. Please refer to Section 4.3 for specific configuration details.

| Method | ZsRE | | | FEVER | | | WikiBigEdit | | | | |
|---|---|---|---|---|---|---|---|---|---|---|---|
| | Eff. | Gen. | Spe. | Eff. | Gen. | Spe. | Eff. | Gen. | Spe. | Personas | Reasoning |
| **GPT-J** | | | | | | | | | | | |
| FT | 15.11 | 13.55 | 2.61 | 14.02 | 13.98 | 9.26 | 21.90 | 17.56 | 8.47 | 13.91 | 9.24 |
| WISE | 34.13 | 33.14 | 26.81 | 93.95 | 92.86 | 57.03 | 49.88 | 45.39 | 30.00 | 40.52 | 21.01 |
| AlphaEdit | 50.23 | 43.31 | 12.54 | 1.89 | 1.87 | 1.85 | 69.66 | 55.03 | 21.20 | 42.60 | 0.07 |
| RLEdit | 73.34 | 68.93 | 22.00 | 14.26 | 13.74 | 14.11 | 66.18 | 60.97 | 32.20 | 55.78 | 25.94 |
| ULTRAEDIT | **78.03** | **72.42** | **27.05** | 97.45 | 96.37 | 79.72 | **73.84** | **66.57** | 37.17 | **56.90** | **29.27** |
| ULTRAEDIT* | 72.95 | 68.68 | 25.91 | **97.89** | **96.73** | **79.85** | 66.46 | 60.54 | **47.90** | 51.73 | – |
| Δ | +4.69 | +3.49 | +0.24 | +3.94 | +3.87 | +22.82 | +4.18 | +5.60 | +15.70 | +1.12 | +3.33 |
| **Mistral-7B-v0.3** | | | | | | | | | | | |
| FT | 13.69 | 12.43 | 19.87 | 23.80 | 23.37 | 16.30 | 13.77 | 14.86 | 11.84 | 11.55 | 7.51 |
| WISE | 34.01 | 32.61 | 46.05 | 81.95 | 76.94 | 40.37 | 37.31 | 33.44 | 11.61 | 27.44 | 8.95 |
| AlphaEdit | 0.00 | 0.00 | 0.00 | 0.00 | 0.00 | 0.00 | 0.00 | 0.00 | 0.00 | 0.00 | 0.00 |
| RLEdit | 72.57 | 68.87 | 23.17 | 92.35 | 91.39 | 71.85 | 57.55 | 52.47 | 28.78 | 49.21 | 22.41 |
| ULTRAEDIT | **85.30** | **80.80** | 47.38 | 97.87 | 96.09 | **84.29** | **76.00** | **70.15** | 46.09 | **62.27** | **35.80** |
| ULTRAEDIT* | 81.13 | 76.78 | **48.06** | **98.23** | **96.97** | 83.43 | 71.78 | 65.63 | **55.40** | 56.11 | – |
| Δ | +12.73 | +11.93 | +2.01 | +5.88 | +5.58 | +12.44 | +18.45 | +17.68 | +26.62 | +13.06 | +13.39 |
| **LLaMA-3-8B-Instruct** | | | | | | | | | | | |
| FT | 12.24 | 10.97 | 9.06 | 16.21 | 13.08 | 5.01 | 13.00 | 11.70 | 6.75 | 11.02 | 4.38 |
| WISE | 40.94 | 40.27 | 37.40 | 86.38 | 86.36 | **72.08** | 34.07 | 32.26 | 28.91 | 29.55 | 21.19 |
| AlphaEdit | 74.34 | 67.85 | 22.94 | 6.39 | 6.14 | 2.72 | 63.24 | 54.68 | 20.17 | 42.58 | 0.01 |
| RLEdit | **91.34** | **89.68** | 41.94 | 94.03 | 90.67 | 68.71 | 75.35 | 70.00 | 37.21 | 65.55 | 28.13 |
| ULTRAEDIT | 90.07 | 87.36 | **49.51** | 95.39 | 91.93 | 67.14 | **79.60** | **73.49** | 48.51 | **66.55** | **35.64** |
| ULTRAEDIT* | 87.80 | 85.48 | 46.74 | **97.18** | **94.64** | 68.62 | 68.99 | 63.59 | **52.28** | 55.04 | – |
| Δ | -1.27 | -2.32 | +7.57 | +3.15 | +3.97 | -3.46 | +4.25 | +3.49 | +15.07 | +1.00 | +7.51 |

To support evaluation across key dimensions, the dataset is divided into three sample types:

- **Editing instances** require the subject entity to appear in the question. This design aligns with the assumptions of many subject-dependent editing methods, which rely on identifying subject-centric representations to apply updates. While ULTRAEDIT does not require this constraint, we include it to ensure compatibility with prior paradigms and enable consistent evaluation of *Efficacy*.
- **Equivalent instances** are paraphrased variants of the editing instances that share the same answers. They evaluate *Generalization*, which measures whether the edit transfers to semantically equivalent rephrasings. We do not enforce whether the subject entity appears in the question.
- **Unrelated instances** contain questions unrelated to the editing fact and are used to assess *Specificity*, that is, whether unrelated knowledge remains unaffected after editing. We do not enforce whether the subject entity appears in the question.

For all three types, the answer is explicitly excluded from the question to prevent lexical leakage and ensure that models must rely on internal knowledge rather than surface patterns. ULTRAEDITBENCH comprises over 2 million complete editing pairs, each containing an editing instance, an equivalent instance, and an unrelated instance. This construction enables ULTRAEDITBENCH to serve as a comprehensive and controlled benchmark for evaluating the precision, generalization, and safety of lifelong model editing methods. And for more details, please refer to Appendix B.1. The full dataset is available at `https://huggingface.co/datasets/XiaojieGu/UltraEditBench`.

Table 3: Results on UltraEditBench across four models.

| Method | GPT-J | | | Mistral-7B-v0.3 | | | LLaMA-3-8B-Instruct | | | Qwen2.5-7B-Instruct | | |
|---|---|---|---|---|---|---|---|---|---|---|---|---|
| | Eff. | Gen. | Spe. | Eff. | Gen. | Spe. | Eff. | Gen. | Spe. | Eff. | Gen. | Spe. |
| FT | 22.03 | 17.61 | 19.60 | 0.06 | 0.36 | 0.07 | 13.50 | 11.92 | 10.18 | 13.20 | 10.53 | 10.27 |
| WISE | 52.51 | 47.88 | 47.50 | 47.21 | 44.55 | 39.13 | 42.27 | 41.65 | 39.79 | – | – | – |
| AlphaEdit | 22.19 | 12.09 | 6.88 | 0.00 | 0.00 | 0.00 | 4.51 | 3.16 | 2.78 | 17.44 | 8.01 | 5.42 |
| RLEdit | 81.42 | 75.35 | 62.13 | 76.50 | 70.68 | 61.29 | 85.69 | **81.88** | 65.64 | 47.08 | 38.76 | 39.27 |
| UltraEdit | **84.03** | 76.62 | 64.03 | **83.71** | **77.30** | 67.26 | **85.70** | 81.28 | 68.73 | 79.01 | 71.45 | 64.10 |
| UltraEdit* | 81.65 | **76.80** | **76.44** | 81.70 | 77.25 | **77.09** | 83.45 | 79.11 | **78.05** | 80.70 | **75.78** | **76.01** |
| Δ | +2.61 | +1.45 | +14.31 | +7.21 | +6.62 | +15.80 | +0.01 | -0.60 | +12.41 | +33.62 | +37.02 | +36.44 |

## 4.2 Experiment Setup

**Dataset & Model** We evaluate the effectiveness and scalability of UltraEdit across five model editing benchmarks: ZsRE, FEVER, WikiBigEdit, UnKE (unstructured long-text data) and our newly constructed UltraEditBench, on a diverse set of open-source models, including GPT-J, Mistral-7B-v0.3, LLaMA-3-8B-Instruct, Qwen2.5-7B-Instruct, Phi-4-14B, and Gemma-3-27B-it. Detailed descriptions of each dataset, along with their corresponding metrics, are provided in Appendix B.1.

**Baseline** We compare UltraEdit against a comprehensive set of baselines, including Finetune (FT), WISE (Wang et al., 2024b), AlphaEdit (Fang et al., 2025), RLEdit (Li et al., 2025), among others. We exclude AnyEdit (Jiang et al., 2025) since it is not designed for lifelong editing. Following (Fang et al., 2025; Li et al., 2025), all methods are evaluated under a consistent setting where each turn includes 100 samples, except for WISE, which is specifically designed to edit one sample per turn. For evaluation protocols, we follow mainstream methods (Fang et al., 2025; Li et al., 2025) by adopting **Exact Match** as the primary evaluation metric. In addition, we employ **WILD (LLM-as-judge)** (Yang et al., 2025) as a complementary evaluation. Further clarifications regarding the evaluation protocols are provided in the Appendix B.3. Comprehensive information on all baseline methods is provided in Appendix B.2 and hyperparameter configurations are presented in Appendix B.4.

## 4.3 Overall Results

We report the performance of UltraEdit alongside the strongest representative methods from each editing paradigm across multiple benchmarks and models. As shown in Table 2 and 3, UltraEdit* is evaluated on 100K edits for ZsRE and FEVER, 500K for WikiBigEdit, and 2M for UltraEditBench, whereas all other methods are evaluated on 20K edits for ZsRE, FEVER, and UltraEditBench, and 17K for WikiBigEdit. UltraEdit consistently leads across standard metrics including *Efficacy*, *Generalization*, and *Specificity*, as well as two newly introduced metrics, *Personas* and *Multi-hop Reasoning*, outperforming all baselines in nearly every setting. This holds true under both the Exact Match and LLM-as-judge evaluation frameworks, for both instruct-tuned and base models, and across diverse data structures of real-world editing instances. Full results for all baseline methods are provided in Appendix C.1. Due to the fact that most existing methods are **several hundred times slower** than UltraEdit, and some require additional training data, scaling them to larger numbers of edits is computationally impractical. Figure 1(b) shows that existing methods degrade as the number of edits grows, indicating poor scalability to ultra-large editing. In contrast, UltraEdit remains robust and stable, sustaining millions of updates in lifelong settings. For instance, on LLaMA-3-8B-Instruct with ZsRE, it maintains near-standard performance across all metrics with only a slight drop under a 5× larger edit load. These results confirm that UltraEdit combines high-quality knowledge injection with long-term stability, providing a practical and scalable solution for real-world model editing. Further discussion of failure scenarios can be found in Section C.4 in Appendix.

# 5 Analysis

## 5.1 Ablation Study

To assess the contribution of each component in ULTRAEDIT, we conduct an ablation study on 20K editing instances from the ZsRE dataset , with results averaged across three backbone models as shown in Table 4. When the lifelong normalization mechanism is entirely removed, we observe a drastic drop in both efficacy and generalization, underscoring the importance of aligning feature distributions across modules during editing. To further assess its effectiveness, we apply lifelong normalization to a randomly selected 25%, 50%, and 75% subset of the editable modules. The results show a clear upward trend in performance with increasing normalization coverage, indicating cumulative and global benefits. Freezing the normalization statistics by disabling online updates during editing causes a catastrophic collapse in all metrics, demonstrating that dynamic calibration of activation statistics is essential for stability and consistent editing across batches.

In addition, we ablate the original norm-based scaling coefficient $\|\tilde{k}_i\|^2$ by replacing it with a direction-based inner product $k_i \cdot \tilde{k}_i$. While both schemes involve inner products, our norm-based formulation more faithfully captures per-sample saliency, whereas the alternative mixes magnitude and alignment, resulting in less stable modulation across edits. Overall, the ablation results confirm that both lifelong normalization and its dynamic updating mechanism are critical for ensuring stable, accurate, and ultra-scalable editing.

Table 4: Ablation study of ULTRAEDIT. Blue numbers indicate a decrease, while Red numbers indicate an increase compared to the full method.

| Variant | Efficacy | Generalization | Specificity |
|---------|----------|----------------|-------------|
| Original | **84.47** | **80.19** | 41.31 |
| w/o lifelong normalization | 36.15↓ 48.32 | 35.25↓ 44.94 | 38.14↓ 3.17 |
| w/ 25% module norm | 77.38↓ 7.09 | 72.68↓ 7.51 | **42.40**↑ 1.09 |
| w/ 50% module norm | 80.81↓ 3.66 | 76.07↓ 4.12 | 42.37↑ 1.06 |
| w/ 75% module norm | 83.64↓ 0.83 | 79.22↓ 0.97 | 41.64↑ 0.33 |
| w/o normalization update | 1.11↓ 83.36 | 1.02↓ 79.17 | 0.07↓ 41.24 |
| w/ $k \cdot \tilde{k}$ | 74.27↓ 10.20 | 70.23↓ 9.96 | 42.18↑ 0.87 |

## 5.2 Lifelong Scalability

ULTRAEDIT is designed not only for editing accuracy but also for long-term scalability. As shown in Figure 1(b) and Figure 4, it maintains strong performance across three key metrics—even as the number of edits increases. As shown in Figure 1(b) and Figure 4, it maintains strong performance across three key metrics as the number of edits increases. This robustness comes from a simple yet effective lifelong normalization strategy that dynamically calibrates internal feature distributions as the model evolves. Unlike many existing methods that degrade much earlier, ULTRAEDIT exhibits progressive stabilization: as edits accumulate, its normalization mechanism regularizes the feature space and improves performance until reaching saturation. By continually adapting to the model's current state without requiring retraining, ULTRAEDIT is particularly well-suited for real-world, lifelong editing scenarios that demand frequent and ongoing updates. Additionally, ULTRAEDIT scales effectively to models with substantially larger parameter sizes. It preserves editing accuracy on both standard language models such as Phi-4-14B and more complex multimodal models like Gemma-3-27B, as shown in Figure 5. And full scalability evaluation is provided in Appendix C.2.

## 5.3 Impact on the General Ability of Post-Edited Models

We evaluate how lifelong editing affects the general ability of post-edited models on four representative benchmarks: SST (Socher et al., 2013), MMLU (Hendrycks et al., 2021), MRPC (Dolan & Brockett, 2005), and NLI (Williams et al., 2018). Results for the original unedited model (Vanilla), AlphaEdit, WISE, RLEdit, and ULTRAEDIT are shown in Figure 6. We observe that as the number of edits increases, methods such as AlphaEdit and finetuning significantly degrade the general ability of post-edited models across all benchmarks, suggesting cumulative interference in the lifelong setting. WISE shows relatively stable performance by storing edits in an external memory component, trading additional memory for reduced interference with the base model. RLEdit causes notable degradation on NLI but remains more stable on other benchmarks. In contrast,

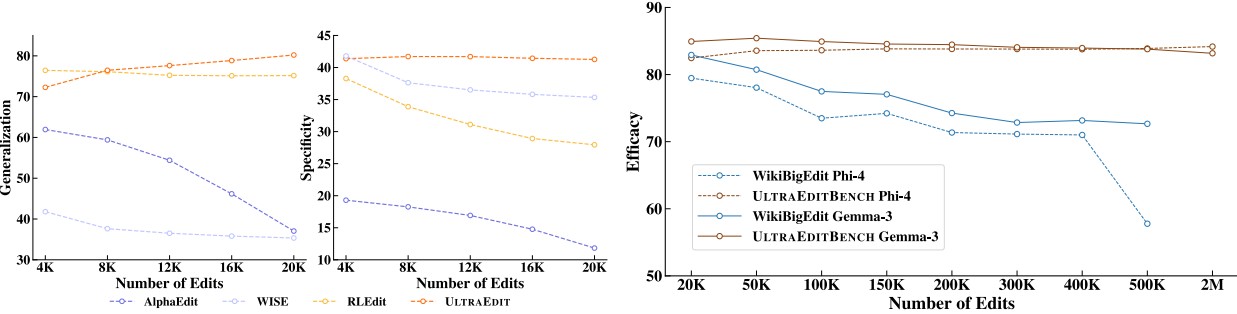

Figure 4: Variation in average generalization and specificity as edits accumulate.

Figure 5: Efficacy of lifelong editing on Phi-4-14B and Gemma-3-27B.

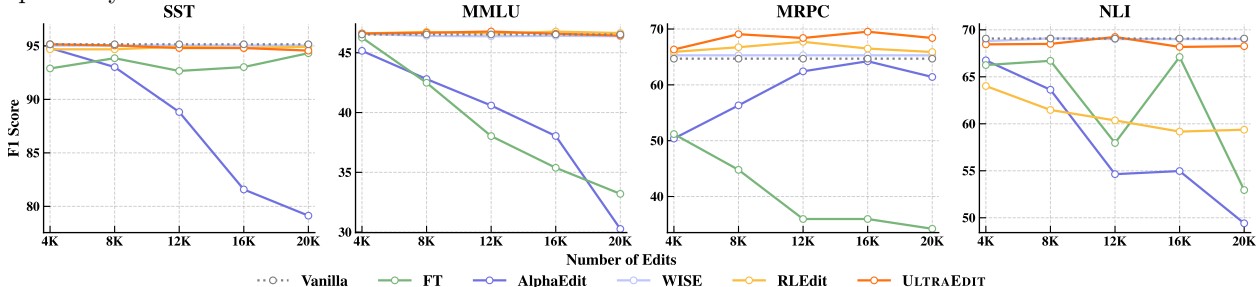

Figure 6: Performance of post-edited LLaMA-3 (20K ZsRE edits) across various benchmarks.

ULTRAEDIT consistently preserves the model's general abilities even after 20K edits, showing almost no deviation from the vanilla baseline across tasks. These findings confirm that ULTRAEDIT introduces the least interference to general capabilities and does not increase the risk of hallucinations, making it well-suited for lifelong model editing. In addition, ULTRAEDIT leads to improved performance on MRPC. We attribute this gain to *Lifelong Normalization* acting as a beneficial regularizer. Theoretically, by dynamically calibrating the first- and second-order statistics $(\mu, \sigma)$ of hidden states, our method prevents the feature distribution from deviating off the pre-trained manifold, thereby mitigating representation collapse often caused by cumulative parameter shifts. Furthermore, since robust editing requires the model to generalize across paraphrased inputs (Eq.(9)), the optimization implicitly enforces semantic invariance by minimizing the distance between representations of semantically equivalent sentences. This objective is well aligned with the MRPC task (paraphrase detection), sharpening the model's discriminative boundary for semantic similarity. For a detailed comparison with other baseline methods, as well as comprehensive description of four benchmarks, please refer to Appendix C.3.

# 6 Conclusion

We present ULTRAEDIT, a fast, stable, and scalable approach to lifelong model editing without additional training, subject reliance, or external memory. Through lifelong normalization, it adapts to evolving model states while maintaining high precision across editing turns. Experiments show that ULTRAEDIT achieves over 7× faster editing with less than one-fourth the VRAM of prior methods, and supports up to 2M edits with stable performance. These efficiency gains make lifelong editing feasible at ultra-large scale and broadly accessible, lowering barriers and enabling wider community participation. To support further research, we release ULTRAEDITBENCH, the largest benchmark to date, with over 2M editing pairs for evaluating ultra-scale, lifelong scenarios.

# 7 Limitation

Owing to limited computational resources and the scale of certain datasets, we were unable to scale all baselines to ultra-large lifelong editing settings or to models with substantially larger parameter counts. Furthermore, the use of LLM-as-a-judge incurs considerable API expenses, so we restricted such evaluations to

the ZsRE dataset. As a result, we did not evaluate the performance of post-edited models under larger-scale conditions (e.g., after 2M edits).

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

# Appendix

The appendix includes the following sections:

- **Section A: Statements.**

- **Section B: Implementation Details of Experiment.**

- **Section C: Full Experiment Results.**

- **Section D: Case Study.**

## A  Statement

### A.1  Real-World Lifelong Application Statement

ULTRAEDIT strictly adheres to a lifelong editing setting, where updates arrive sequentially and **NO** future edits are known in advance. The proposed lifelong normalization mechanism does not need to aggregate statistics across all editing turns; instead, it incrementally accumulates feature statistics from previously observed samples up to the current turn using a causal running average, without ever accessing future data. Importantly, each editing turn only requires the statistics from the last turn, which are then updated iteratively, rather than storing or recomputing information from all past samples. As detailed in equation 1–equation 3 in Section 3.2, the normalization statistics $\mu$ and $\sigma$ are computed in a running fashion and updated after each editing turn based on observed feature vectors.

### A.2  Ethical Considerations

While ULTRAEDIT democratizes the ability to maintain LLMs on consumer hardware, this efficiency also lowers the barrier for malicious actors to inject misinformation or compromise safety alignment at scale. To mitigate these risks, we recommend adopting Integrity Verification mechanisms, such as cryptographic hashing (e.g., SHA-256) and digital signatures, to ensure model provenance and detect unauthorized alterations. Furthermore, we advocate for Defensive Editing, where the efficiency of ULTRAEDIT is leveraged to rapidly patch discovered vulnerabilities or reverse malicious injections, thereby serving as a countermeasure against model poisoning.

# B  Implementation Details

## B.1  Dataset and Metrics

**ZsRE** (Zero-shot Relation Extraction) dataset (Levy et al., 2017) is a widely adopted benchmark for evaluating factual consistency and knowledge editing in language models. Each example in the dataset comprises a question–answer pair that encapsulates a factual relation. To facilitate comprehensive assessment of model editing capabilities, the dataset is augmented with two types of auxiliary samples: (1) paraphrased variants of the original question, which test the model's ability to generalize the update, and (2) unrelated but structurally similar questions, which assess the specificity of the edit by ensuring unrelated knowledge remains unaffected. This design enables precise evaluation of editing accuracy, generalization to rephrased inputs, and the preservation of unrelated facts, making ZsRE particularly well-suited for controlled knowledge update tasks. The dataset contains a total of 178,196 examples.

Following prior work (Meng et al., 2023; Fang et al., 2025; Li et al., 2025), we evaluate various model editing approaches on ZsRE using standard metrics. Given a large language model $f_\theta$, a target editing pair $(x^e, y^e)$, its paraphrased equivalent $x^{e'}$, and a set of unrelated knowledge pairs $(x^u, y^u)$, we assess the following three metrics:

*Efficacy (Eff.)* measures whether the model correctly incorporates the edit by verifying if the top-1 prediction for the edited input $x^e$ matches the target label $y^e$:

$$\mathbb{E}\left\{y^e = \arg\max_{y'} \mathbb{P}_{f_\theta}(y' \mid x^e)\right\}. \tag{8}$$

*Generalization (Gen.)* evaluates whether the model successfully generalizes the edit to paraphrased forms of the input, by checking if the top-1 prediction for $x^{e'}$ remains consistent with $y^e$:

$$\mathbb{E}\left\{y^e = \arg\max_{y'} \mathbb{P}_{f_\theta}(y' \mid x^{e'})\right\}. \tag{9}$$

*Specificity (Spe.)* assesses the model's ability to retain unrelated knowledge by ensuring the top-1 prediction for each unrelated input $x^u$ continues to match its original label $y^u$:

$$\mathbb{E}\left\{y^u = \arg\max_{y'} \mathbb{P}_{f_\theta}(y' \mid x^u)\right\}. \tag{10}$$

**FEVER** (Fact Extraction and VERification) dataset (Thorne et al., 2018) is a large-scale benchmark designed for evaluating factual consistency and claim verification in natural language. Each example comprises a natural language claim accompanied by a label (Supported, Refuted, or Not Enough Information) determined based on evidence retrieved from Wikipedia. The claims are either directly extracted from Wikipedia or are semantically modified versions of actual content, while the supporting evidence may span multiple sentences or even multiple documents. This setup enables fine-grained assessment of a model's ability to confirm, reject, or abstain from factual assertions. The dataset contains a total of 114,422 examples. For consistency in evaluation, we adopt the same metric definitions used in ZsRE.

**WikiBigEdit** (Thede et al., 2025) is a large-scale benchmark designed for lifelong knowledge editing. The dataset contains a total of 506,035 editing pairs, all derived from real-world Wikidata revisions collected across eight time steps over a six-month period. To support comprehensive evaluation, WikiBigEdit defines five core metrics: *Update*, *Rephrase*, *Locality*, *Personas*, and *Multi-hop Reasoning*. Among them, *Update*, *Rephrase*, and *Locality* correspond closely to the standard editing criteria of *Efficacy*, *Generalization*, and *Specificity*, and are evaluated using the same methodology as in ZsRE. *Personas* and *Multi-hop Reasoning* extend the evaluation scope to identity conditioning and complex reasoning, respectively. A total of 490,519 examples support the first four metrics, while 17,541 are specifically annotated for multi-hop reasoning, enabling focused evaluation of a model's ability to handle compositional queries and long-range dependencies. Reflecting realistic and evolving knowledge dynamics, WikiBigEdit supports iterative assessment of a model's capacity to incorporate

factual updates over time. Constructed via an automated data pipeline, the benchmark is continuously expandable as new Wikidata edits become available. Specifically, *Personas* evaluates whether the model correctly answers identity-conditioned prompts $x^p$, while *Multi-hop Reasoning* measures the model's ability to resolve compositional or chained queries $x^m$. Their accuracy is computed as:

$$\text{Personas:} \quad \mathbb{E}\left\{y^p = \arg\max_{y'} \mathbb{P}_{f_\theta}(y' \mid x^p)\right\}, \tag{11}$$

$$\text{Multi-hop Reasoning:} \quad \mathbb{E}\left\{y^m = \arg\max_{y'} \mathbb{P}_{f_\theta}(y' \mid x^m)\right\}. \tag{12}$$

ULTRAEDITBENCH comprises 2,008,326 editing pairs and adheres to the model editing evaluation framework established in ZsRE. Inference results on different models across all datasets, including ULTRAEDITBENCH are shown in Table 5. The results indicate that the knowledge in ULTRAEDITBENCH aligns well with the requirements of model editing tasks. The diversity verification of ULTRAEDITBENCH is presented in Tables 6, 7, and 8. The average prompt length is 11.29 tokens per sentence, and the Hapax Legomena Ratio reaches 56.45%. Together, these results demonstrate the overall diversity of the dataset.

Table 5: Inference results on pre-edited models. ZsRE, FEVER, and ULTRAEDITBENCH use 20K edits, while WikiBigEdit uses 17K.

| Model | ZsRE | | | FEVER | | | WikiBigEdit | | | | | ULTRAEDITBENCH | | |
|---|---|---|---|---|---|---|---|---|---|---|---|---|---|---|
| | Eff. | Gen. | Spe. | Eff. | Gen. | Spe. | Eff. | Gen. | Spe. | Personas | Reasoning | Eff. | Gen. | Spe. |
| GPT-J | 27.22 | 26.42 | 27.33 | 9.61 | 9.68 | 15.90 | 29.97 | 29.08 | 32.58 | 26.46 | 21.81 | 22.01 | 21.47 | 22.20 |
| Mistral-7B-v0.3 | 44.46 | 43.55 | 48.08 | 0.41 | 0.50 | 1.98 | 39.14 | 38.21 | 41.62 | 35.54 | 29.93 | 30.83 | 29.94 | 31.11 |
| LLaMA-3-8B-Instruct | 36.76 | 35.83 | 38.93 | 0.02 | 0.02 | 0.26 | 24.92 | 35.46 | 38.87 | 32.70 | 26.42 | 27.29 | 26.40 | 27.24 |
| Qwen2.5-7B-Instruct | 34.32 | 33.39 | 38.06 | 0.57 | 0.60 | 2.17 | 30.97 | 30.40 | 34.50 | 28.43 | 22.09 | 26.03 | 25.22 | 26.35 |

Table 6: Domain diversity of ULTRAEDITBENCH

| Type | Person | Organization | Geography | History | Society | Technology | Arts | Politcs | Culture | Others |
|---|---|---|---|---|---|---|---|---|---|---|
| Rate | 45.13% | 13.21% | 6.67% | 2.88% | 1.27% | 0.25% | 0.17% | 0.05% | 0.01% | 30.36% |

Table 7: Languages of subjects in ULTRAEDITBENCH.

| Type | English | German | Italian | French | Indonesian | Spanish | Tagalog/Filipino | Welsh | Finnish | Dutch | Others |
|---|---|---|---|---|---|---|---|---|---|---|---|
| Rate | 31.78% | 7.82% | 5.74% | 4.52% | 4.16% | 3.46% | 2.75% | 2.57% | 2.53% | 2.36% | 32.30% |

Table 8: Answer length in ULTRAEDITBENCH.

| Length | 1 | 2 | 3 | 4 | 5 | 6 | 7 | 8 | 9 | 10 |
|---|---|---|---|---|---|---|---|---|---|---|
| Rate | 33.66% | 36.53% | 17.85% | 6.12% | 3.38% | 1.27% | 0.55% | 0.31% | 0.16% | 0.07% |

**UnKE** (Deng et al., 2025) consists of 1,000 unstructured long-text samples, but it only provides editing instances and equivalent instances. To construct a more comprehensive benchmark, we take another 1,000 samples from the long-text dataset AKEW (Wu et al., 2024b) as unrelated instances and combine them with UnKE, forming the final UnKE dataset. In addition to the standard metrics of efficacy, generalization, and specificity, we also adopt *BERTScore* and *ROUGE-L* from UnKE as two additional evaluation metrics. *BERTScore* measures the semantic similarity between the generated output $\hat{y}^e$ and the target $y^e$ by computing token-level cosine similarities in the embedding space:

$$\text{BERTScore}(y^e, \hat{y}^e) = \frac{1}{2}\left(\frac{1}{|\hat{y}^e|}\sum_{c\in\hat{y}^e}\max_{r\in y^e}\cos(c,r) \; + \; \frac{1}{|y^e|}\sum_{r\in y^e}\max_{c\in\hat{y}^e}\cos(r,c)\right). \tag{13}$$

$c$ and $r$ denote token embeddings from the generated output $\hat{y}^e$ and the target $y^e$, respectively. For each token $c$ in $\hat{y}^e$, we find the most similar token $r$ in $y^e$ (by cosine similarity), and vice versa. The final score averages these two directions, capturing both precision- and recall-oriented matching.

*ROUGE-L* evaluates the longest common subsequence (LCS) between the generated output $\hat{y}^e$ and the target $y^e$. Let $LCS(y^e, \hat{y}^e)$ be their longest common subsequence length, then:

$$P_{LCS} = \frac{LCS(y^e, \hat{y}^e)}{|\hat{y}^e|}, \quad R_{LCS} = \frac{LCS(y^e, \hat{y}^e)}{|y^e|}, \tag{14}$$

$$\text{ROUGE-L}(y^e, \hat{y}^e) = \frac{(1 + \beta^2) \cdot P_{LCS} \cdot R_{LCS}}{R_{LCS} + \beta^2 \cdot P_{LCS}}, \quad \beta = 1. \tag{15}$$

Datasets such as Counterfact (Meng et al., 2022), MQuAKE (Zhong et al., 2023a), KnowEdit (Zhang et al., 2024), and QAEdit (Yang et al., 2025) are excluded from our evaluation, due to their limited scale to support meaningful large-scale lifelong model editing.

We follow the LLM-as-judge template from (Yang et al., 2025), as shown in Table 9. For evaluation, we employ DeepSeek-V3.1 (Non-thinking Mode) (Liu et al., 2024) as the third-party model, with outputs truncated to 512 tokens.

Table 9: LLM-as-judge template

Your job is to look at a question, a gold target, and a predicted answer, and then assign a
↪ grade of either ["CORRECT", "INCORRECT"].

The following are examples of CORRECT predicted answers.
```
Question: What are the names of Barack Obama's children?
Gold target: Malia Obama and Sasha Obama
Predicted answer 1: sasha and malia obama
Predicted answer 2: Malia and Sasha Obama are the names of Barack Obama's children.
```
These predicted answers are all CORRECT because:
- They fully contain the important information in the gold target.
- They do not contain any information that contradicts the gold target.

The following are examples of INCORRECT predicted answers.
```
Question: What are the names of Barack Obama's children?
Gold target: Malia and Sasha
Predicted answer 1: Malia.
Predicted answer 2: Malia, Sasha, and Susan.
Predicted answer 3: Malia and Sasha, Malia and Sasha, Malia and Sasha, Malia and Sasha
↪ (repeated answer)
```
These predicted answers are all INCORRECT because:
- A factual statement in the answer contradicts the gold target or contain repeated answer.

Here is a sample. Simply reply with either CORRECT or INCORRECT.

```
Question: {question}
Gold target: {target}
Predicted answer: {predicted_answer}
```

According to the gold target, please grade the predicted answer of this question as one of:
A: CORRECT
B: INCORRECT

Just return the letters "A" or "B", with no text around it.

## B.2 Description of Baselines

**FT (Fine-Tuning)** in model editing refers to the process of updating a pre-trained language model's parameters by optimizing them on a small set of new data containing the desired knowledge. This approach aims to adjust the model's behavior to reflect updated or corrected information without retraining the model from scratch. In the context of model editing, fine-tuning typically involves minimizing a loss function such as cross-entropy on carefully selected inputs and targets related to the editing fact, while optionally applying regularization to avoid catastrophic forgetting. Despite its simplicity and effectiveness, FT can lead to unintended changes in the model's general abilities or interfere with unrelated knowledge, especially when used repeatedly or with large learning rates.

**MEND** enables efficient post-hoc editing of large pre-trained models using only a single input-output pair by training lightweight auxiliary networks that transform standard fine-tuning gradients into localized, low-rank parameter updates. MEND leverages the inherent rank-1 structure of gradients in neural networks to map input activations and output deltas through a small MLP, producing edits that are reliable, local, generalizable, and scalable to models with over 10 billion parameters, all without retraining or modifying the original model's predictions on unrelated inputs.

**MEMIT** modifies factual associations in language models by identifying and updating critical MLP layers responsible for knowledge recall, computing target hidden states for new facts, and applying analytically derived weight updates that distribute edits across multiple layers. The method treats MLPs as linear associative memories and performs batch updates to insert new associations while preserving previously stored information, enabling efficient memory editing within the model's internal structure.

**MALMEN** is a scalable hypernetwork-based method for editing large language models, which addresses the limitations of MEND by formulating parameter shift aggregation as a least squares problem solved via the normal equation to prevent cancellation effects, and decoupling the training of the hyper-network and language model to support arbitrary batch sizes, enabling efficient and memory-economic editing of thousands of facts while maintaining strong locality and generalization.

**RECT** introduces a regularization-based approach to model editing that constrains the complexity of edit updates by preserving only the top-k% of weight changes based on their relative change magnitude, thereby mitigating overfitting and protecting the model's general abilities. By identifying and retaining the most impactful parameter updates while zeroing out less significant ones, RECT reduces interference with original model knowledge and avoids degradation across downstream tasks. This method ensures that the edited model maintains a balance between incorporating new factual information and preserving its overall performance.

**WISE** proposes a dual-parametric memory approach for lifelong model editing, where pretrained knowledge is kept in a main memory and all edits are stored in a separate side memory, which is a copy of a Transformer layer's value matrix. To determine which memory to use at inference, WISE employs a routing mechanism based on activation differences, directing queries to side memory if they relate to edited knowledge. To handle continuous edits without conflict, WISE introduces a knowledge sharding technique that stores different edits in randomly masked subspaces of the side memory, and then merges these using Ties-Merge to maintain consistency. This design allows WISE to achieve high reliability, generalization, and locality simultaneously, overcoming the limitations of traditional long-term or working memory editing methods.

**PRUNE** introduces a plug-and-play framework for sequential model editing by restraining the condition number of the edited matrix, which is identified as the main factor causing degradation in general abilities during repeated edits. It operates by analyzing the singular value decomposition of accumulated edit updates and selectively reducing excessively large singular values through a restraint function, thereby minimizing perturbations to the model's original knowledge associations while preserving the newly injected information. This method reduces overfitting from edit accumulation and maintains model stability without interfering with editing efficacy, making it compatible with various existing editing approaches.

**AlphaEdit** introduces a null-space constrained knowledge editing approach by projecting parameter perturbations onto the null space of preserved knowledge, ensuring updates do not interfere with existing information. This projection allows the editing process to focus solely on updating target knowledge without trade-offs, eliminating the need for additional constraints in the optimization objective. The method achieves substantial

improvements in editing performance and generalization with minimal implementation overhead, maintaining the integrity of preserved knowledge and the model's overall capabilities during sequential edits.

**RLEdit** formulates lifelong model editing as a reinforcement learning problem by treating the hypernetwork as an agent that generates parameter updates based on the current state of the language model and input knowledge, where the update is considered an action and the editing performance defines the reward. It trains the hypernetwork offline using accumulated rewards across a sequence of edits, incorporating a reward function that balances knowledge injection, preservation of unrelated information, memory backtracking for prior edits, and regularization to constrain update magnitude. This design enables the hypernetwork to adaptively produce edits that are compatible with dynamically evolving model parameters over long editing trajectories.

## B.3 Evaluation Statement

We evaluate the model after completing all editing turns and compute the metrics on the entire set of edited samples at once, rather than reporting per-turn performance. This evaluation protocol ensures that the reported results reflect the final performance of the model after lifelong editing, rather than the transient performance at individual turns. Consequently, our evaluation setting is both fair and consistent with the true goal of lifelong editing.

### B.4 Hyperparameter Setting

All experiments are conducted on a single NVIDIA A800 GPU, and ULTRAEDIT introduces only two hyperparameters: the learning rate $\eta$ and the editable module. In all settings, $\eta$ is set to 1e-6. Regarding the selection of editable modules, we strictly follow the parameter settings used in RLEdit for fair comparison. For WikiBigEdit and ULTRAEDITBENCH, which are not used in their work, the editable modules are aligned with those used for ZsRE. The details of the editable modules are shown in Table 10, where the numbers indicate the corresponding layer indices.

Table 10: Editable Module settings of ULTRAEDIT across different models and datasets

| Dataset | Model | Editable Module |
|---|---|---|
| ZsRE | GPT-J | [18-26].mlp.fc_out |
| | Mistral-7B-v0.3 | [29, 30].mlp.down_proj |
| | LLaMA-3-8B-Instruct | [11-15].mlp.gate_proj, [18-24].mlp.up_proj |
| | Qwen2.5-7B-Instruct | [18-26].mlp.gate_proj,[18-26].mlp.up_proj |
| FEVER | GPT-J | [25,26].mlp.fc_out |
| | Mistral-7B-v0.3 | [29, 30].mlp.down_proj |
| | LLaMA-3-8B-Instruct | [22-30].mlp.gate_proj, [22-30].mlp.up_proj |
| | Qwen2.5-7B-Instruct | [18-26].mlp.gate_proj,[18-26].mlp.up_proj |
| WikiBigEdit | GPT-J | [19-26].mlp.fc_out |
| | Mistral-7B-v0.3 | [29, 30].mlp.down_proj |
| | LLaMA-3-8B-Instruct | [11-15].mlp.gate_proj, [18-24].mlp.up_proj |
| | Qwen2.5-7B-Instruct | [19-26].mlp.gate_proj,[19-26].mlp.up_proj |
| | Phi-4-14B | [30-38].mlp.down_proj |
| | Gemma-3-27B-it | [52-60].mlp.gate_proj,[52-60].mlp.up_proj |
| ULTRAEDITBENCH | GPT-J | [18-26].mlp.fc_out |
| | Mistral-7B-v0.3 | [29, 30].mlp.down_proj |
| | LLaMA-3-8B-Instruct | [11-15].mlp.gate_proj, [18-24].mlp.up_proj |
| | Qwen2.5-7B-Instruct | [18-26].mlp.gate_proj,[18-26].mlp.up_proj |
| | Phi-4-14B | [30-38].mlp.down_proj |
| | Gemma-3-27B-it | [52-60].mlp.gate_proj,[52-60].mlp.up_proj |
| UnKE | GPT-J | [18-26].mlp.fc_out |
| | Mistral-7B-v0.3 | [29, 30].mlp.down_proj |
| | LLaMA-3-8B-Instruct | [11-15].mlp.gate_proj, [18-24].mlp.up_proj |
| | Qwen2.5-7B-Instruct | [18-26].mlp.gate_proj,[18-26].mlp.up_proj |

For methods that require additional training data, such as MEND, MALMEN, RLEdit and WISE, we follow the experimental setup described in RLEdit. Specifically, for the ZsRE and FEVER datasets, the training set for MEND and MALMEN contains the total number of samples excluding the editing examples. For large-scale datasets like WikiBigEdit and ULTRAEDITBENCH, the size of the training set is equal to the number of editing examples. In all RLEdit and WISE experiments, the training and editing set sizes are always matched.

## C Full Experimental Results

### C.1 Extended Baseline Comparison

This section presents a comprehensive comparison of ULTRAEDIT against baseline methods across five datasets and four model architectures. Results on UnKE are reported in Tables 11 and 12, while Table 13 presents the evaluation on zsRE using LLM-as-judge. Detailed results in Tables 14, 15, and 16 further highlight the strong performance of ULTRAEDIT across diverse editing scenarios.

Table 11: Results across GPT-J and Mistral on UnKE. "*" indicates results from editing 1,000 instances; all others are based on 500 instances.

| Method | GPT-J | | | | | Mistral-7B-v0.3 | | | | |
|---|---|---|---|---|---|---|---|---|---|---|
| | Eff. | Gen. | Spe. | Bert Score | Rouge-L | Eff. | Gen. | Spe. | Bert Score | Rouge-L |
| FT | 46.93 | 46.49 | 50.49 | 51.02 | 11.85 | 70.87 | 70.43 | 70.85 | 70.95 | 13.39 |
| WISE | 91.26 | 89.81 | 65.07 | **85.60** | 42.01 | 89.77 | 88.78 | 70.68 | 79.61 | 35.64 |
| AlphaEdit | 75.96 | 63.20 | **77.19** | 78.83 | 35.60 | 81.39 | 76.78 | **78.40** | **82.90** | **39.49** |
| RLEdit | 86.53 | 84.94 | 56.35 | 79.97 | 33.68 | 46.63 | 44.35 | 15.29 | 33.80 | 6.93 |
| ULTRAEDIT | 91.34 | 89.84 | 64.53 | 81.27 | 36.43 | **92.94** | **90.46** | 65.89 | 74.68 | 30.09 |
| ULTRAEDIT* | **92.49** | **91.15** | 65.93 | 83.02 | **43.28** | 62.79 | 61.72 | 38.06 | 46.69 | 7.04 |
| Δ | +1.23 | +1.34 | -11.26 | -2.58 | +1.27 | +3.17 | +1.68 | -12.51 | -8.22 | -9.40 |

Table 12: Results across LLaMA-3 and Qwen2.5 on UnKE.

| Method | LLaMA-3-8B-Instruct | | | | | Qwen2.5-7B-Instruct | | | | |
|---|---|---|---|---|---|---|---|---|---|---|
| | Eff. | Gen. | Spe. | Bert Score | Rouge-L | Eff. | Gen. | Spe. | Bert Score | Rouge-L |
| FT | 67.09 | 66.39 | 67.31 | 64.65 | 6.89 | 45.72 | 45.44 | 32.55 | 47.67 | 12.12 |
| WISE | 83.95 | 82.93 | 58.08 | 80.43 | 32.02 | - | - | - | - | - |
| AlphaEdit | 71.07 | 68.31 | 69.17 | 81.39 | 37.37 | 39.98 | 37.78 | 25.14 | 64.72 | 23.36 |
| RLEdit | 91.91 | 91.03 | 62.23 | 85.76 | **56.78** | **92.65** | **91.59** | 63.17 | **84.03** | **51.68** |
| ULTRAEDIT | 93.05 | 91.43 | 65.62 | 84.29 | 45.33 | 86.39 | 84.33 | 61.11 | 81.14 | 32.22 |
| ULTRAEDIT* | **94.09** | **92.68** | **73.64** | **85.84** | 50.92 | 88.84 | 87.01 | **72.17** | 82.02 | 36.31 |
| Δ | +2.18 | +1.65 | +4.47 | +0.08 | -5.86 | -3.81 | -4.58 | +9.00 | -2.01 | -15.37 |

Table 13: Evaluation on zsRE using LLM-as-judge across three models. We exclude GPT-J due to its limited capability in instruction-shot.

| Method | Mistral-7B-v0.3 | | | LLaMA-3-8B-Instruct | | | Qwen2.5-7B-Instruct | | |
|---|---|---|---|---|---|---|---|---|---|
| | Eff. | Gen. | Spe. | Eff. | Gen. | Spe. | Eff. | Gen. | Spe. |
| FT | 0.18 | 0.22 | 0.14 | 0.03 | 0.02 | 0.00 | 0.43 | 0.29 | 0.10 |
| WISE | 0.90 | 1.03 | 0.13 | 8.90 | 8.40 | 12.23 | - | - | - |
| AlphaEdit | 0.00 | 0.00 | 0.00 | **58.47** | **56.36** | 24.64 | 0.08 | 0.08 | 0.00 |
| RLEdit | 7.75 | 5.50 | 0.00 | 47.09 | 44.31 | 24.46 | 17.11 | 15.67 | 15.52 |
| ULTRAEDIT | **24.49** | **21.97** | **19.95** | 52.85 | 49.77 | **40.13** | **32.65** | **30.45** | **33.60** |
| Δ | +16.74 | +16.47 | +19.81 | -5.62 | -6.59 | +15.49 | +15.54 | +14.78 | +18.08 |

Table 14: Extended results across four models on ULTRAEDITBENCH.

| Method | GPT-J | | | Mistral-7B-v0.3 | | | LLaMA-3-8B-Instruct | | | Qwen2.5-7B-Instruct | | |
|---|---|---|---|---|---|---|---|---|---|---|---|---|
| | Eff. | Gen. | Spe. | Eff. | Gen. | Spe. | Eff. | Gen. | Spe. | Eff. | Gen. | Spe. |
| MEND | 1.71 | 1.71 | 1.83 | 0.00 | 0.00 | 0.00 | 0.00 | 0.00 | 0.00 | 3.48 | 3.45 | 3.43 |
| MEMIT | 0.25 | 0.18 | 0.20 | 0.00 | 0.00 | 0.00 | 0.74 | 0.45 | 0.21 | 0.82 | 0.32 | 0.69 |
| MALMEN | 0.76 | 0.48 | 0.78 | 2.42 | 2.48 | 3.47 | 40.64 | 34.18 | 37.29 | 4.36 | 3.48 | 4.38 |
| RECT | 0.09 | 0.06 | 0.08 | 0.00 | 0.00 | 0.00 | 0.55 | 0.09 | 0.00 | 1.86 | 1.55 | 1.84 |
| PRUNE | 0.32 | 0.27 | 0.28 | 0.00 | 0.00 | 0.00 | 1.21 | 0.85 | 0.32 | 0.27 | 0.06 | 0.14 |
| ULTRAEDIT | **84.03** | 76.62 | 64.03 | **83.71** | **77.30** | 67.26 | **85.70** | **81.28** | 68.73 | 79.01 | 71.45 | 64.10 |
| ULTRAEDIT* | 81.65 | **76.80** | **76.44** | 81.70 | 77.25 | **77.09** | 83.45 | 79.11 | **78.05** | **80.70** | **75.78** | **76.01** |
| Δ | +82.32 | +75.09 | +74.61 | +81.29 | +74.82 | +73.62 | +45.06 | +47.10 | +40.76 | +76.34 | +72.30 | +71.63 |

Table 15: Extended results on three datasets across three models.

| Method | ZsRE | | | FEVER | | | WikiBigEdit | | | | |
|---|---|---|---|---|---|---|---|---|---|---|---|
| | Eff. | Gen. | Spe. | Eff. | Gen. | Spe. | Eff. | Gen. | Spe. | Personas | Reasoning |
| GPT-J | | | | | | | | | | | |
| MEND | 2.52 | 2.55 | 0.19 | 52.80 | 51.44 | 45.42 | 0.02 | 0.01 | 0.02 | 0.02 | 0.02 |
| MEMIT | 0.00 | 0.00 | 0.00 | 5.54 | 5.03 | 5.46 | 1.59 | 1.59 | 0.50 | 0.80 | 0.00 |
| MALMEN | 0.02 | 0.01 | 0.02 | 1.33 | 1.25 | 2.92 | 0.00 | 0.01 | 0.01 | 0.00 | 0.00 |
| RECT | 0.03 | 0.03 | 0.12 | 18.12 | 18.08 | 12.27 | 2.13 | 1.99 | 0.59 | 2.06 | 0.00 |
| PRUNE | 0.00 | 0.00 | 0.01 | 5.25 | 4.72 | 5.20 | 2.36 | 2.32 | 1.01 | 1.86 | 0.00 |
| UltraEdit | **78.03** | **72.42** | **27.05** | 97.45 | 96.37 | 79.72 | **73.84** | **66.57** | 37.17 | **56.90** | **29.27** |
| UltraEdit* | 72.95 | 68.68 | 25.91 | **97.89** | **96.73** | **79.85** | 66.46 | 60.54 | **47.90** | 51.73 | – |
| Δ | +75.51 | +69.87 | +26.86 | +45.09 | +45.29 | +34.43 | +71.48 | +64.25 | +46.89 | +54.84 | +29.25 |
| Mistral-7B-v0.3 | | | | | | | | | | | |
| MEND | 0.00 | 0.00 | 0.00 | 0.00 | 0.00 | 0.00 | 0.01 | 0.00 | 0.00 | 0.01 | 0.00 |
| MEMIT | 0.00 | 0.00 | 0.00 | 0.00 | 0.00 | 0.00 | 0.00 | 0.00 | 0.00 | 0.00 | 0.00 |
| MALMEN | 0.00 | 0.00 | 0.00 | 18.42 | 17.43 | 12.09 | 0.00 | 0.00 | 0.01 | 0.00 | 0.00 |
| RECT | 0.00 | 0.00 | 0.00 | 0.00 | 0.00 | 0.00 | 0.00 | 0.00 | 0.00 | 0.00 | 0.00 |
| PRUNE | 0.00 | 0.00 | 0.00 | 0.00 | 0.00 | 0.00 | 0.00 | 0.00 | 0.00 | 0.00 | 0.00 |
| UltraEdit | **85.30** | **80.80** | 47.38 | 97.87 | 96.09 | **84.29** | **76.00** | **70.15** | 46.09 | **62.27** | **35.80** |
| UltraEdit* | 81.13 | 76.78 | **48.06** | **98.23** | **96.97** | 83.43 | 71.78 | 65.63 | **55.40** | 56.11 | – |
| Δ | +85.30 | +80.80 | +48.06 | +79.81 | +79.54 | +72.20 | +75.99 | +70.15 | +55.39 | +62.26 | +35.80 |
| LLaMA-3-8B-Instruct | | | | | | | | | | | |
| MEND | 0.00 | 0.00 | 0.00 | 36.19 | 36.19 | 24.31 | 0.01 | 0.01 | 0.10 | 0.01 | 0.00 |
| MEMIT | 0.14 | 0.14 | 1.40 | 0.02 | 0.02 | 0.02 | 0.02 | 0.02 | 0.09 | 0.02 | 0.00 |
| MALMEN | 0.20 | 0.12 | 0.09 | 94.50 | 91.26 | 67.76 | 0.00 | 0.00 | 0.00 | 0.00 | 0.00 |
| RECT | 0.00 | 0.00 | 0.00 | 0.01 | 0.00 | 0.00 | 0.21 | 0.24 | 0.06 | 0.29 | 0.00 |
| PRUNE | 0.00 | 0.00 | 0.24 | 0.02 | 0.02 | 0.00 | 0.02 | 0.02 | 0.09 | 0.02 | 0.00 |
| UltraEdit | **90.07** | **87.36** | **49.51** | 95.39 | 91.93 | 67.14 | **79.60** | **73.49** | 48.51 | **66.55** | **35.64** |
| UltraEdit* | 87.80 | 85.48 | 46.74 | **97.18** | **94.64** | **68.62** | 68.99 | 63.59 | **52.28** | 55.04 | – |
| Δ | +89.87 | +87.22 | +48.11 | +2.68 | +3.38 | +0.86 | +79.39 | +73.25 | +52.18 | +66.26 | +35.64 |

Table 16: Results on three datasets across Qwen2.5.

| Method | ZsRE | | | FEVER | | | WikiBigEdit | | | | |
|---|---|---|---|---|---|---|---|---|---|---|---|
| Qwen2.5-7B-Instruct | | | | | | | | | | | |
| | Eff. | Gen. | Spe. | Eff. | Gen. | Spe. | Eff. | Gen. | Spe. | Personas | Reasoning |
| FT | 14.02 | 10.91 | 3.39 | 26.09 | 24.62 | 21.36 | 10.35 | 7.59 | 3.68 | 5.55 | 5.84 |
| MEND | 15.00 | 14.41 | 0.47 | 76.43 | 77.66 | 40.39 | 0.00 | 0.00 | 0.00 | 0.00 | 0.00 |
| MEMIT | 0.02 | 0.02 | 0.17 | 0.08 | 0.12 | 0.15 | 0.54 | 0.33 | 0.38 | 0.32 | 0.00 |
| MALMEN | 0.00 | 0.00 | 0.00 | 0.06 | 0.06 | 0.07 | 0.06 | 0.02 | 0.02 | 0.00 | 0.00 |
| RECT | 0.00 | 0.00 | 0.00 | 3.36 | 3.10 | 3.20 | 2.34 | 2.29 | 0.83 | 0.78 | 0.00 |
| PRUNE | 0.01 | 0.02 | 0.07 | 0.08 | 0.15 | 0.11 | 2.34 | 2.34 | 0.97 | 1.37 | 0.00 |
| AlphaEdit | 16.32 | 13.96 | 1.66 | 32.78 | 31.19 | 22.12 | 20.31 | 15.49 | 2.17 | 9.01 | 0.23 |
| RLEdit | **84.70** | **82.79** | 38.26 | 0.00 | 0.00 | 0.00 | 2.83 | 1.81 | 0.43 | 1.45 | 0.39 |
| UltraEdit | 82.03 | 77.08 | 45.51 | **97.97** | 93.91 | 68.86 | **73.37** | **65.86** | 45.12 | **54.65** | **32.74** |
| UltraEdit* | 78.39 | 74.72 | **49.27** | 97.49 | **94.83** | **68.99** | 66.34 | 60.42 | **51.74** | 50.53 | – |
| Δ | -2.67 | -5.71 | +11.01 | +21.54 | +17.17 | +28.60 | +53.06 | +50.37 | +48.06 | +45.64 | +26.90 |

## C.2   Supplementary Lifelong Scalability Evaluation

Figure 7 shows a comparison of ULTRAEDIT and baseline methods as the number of edits gradually increases. The results demonstrate that ULTRAEDIT maintains stable performance, highlighting its lifelong scalability with respect to the number of edits.

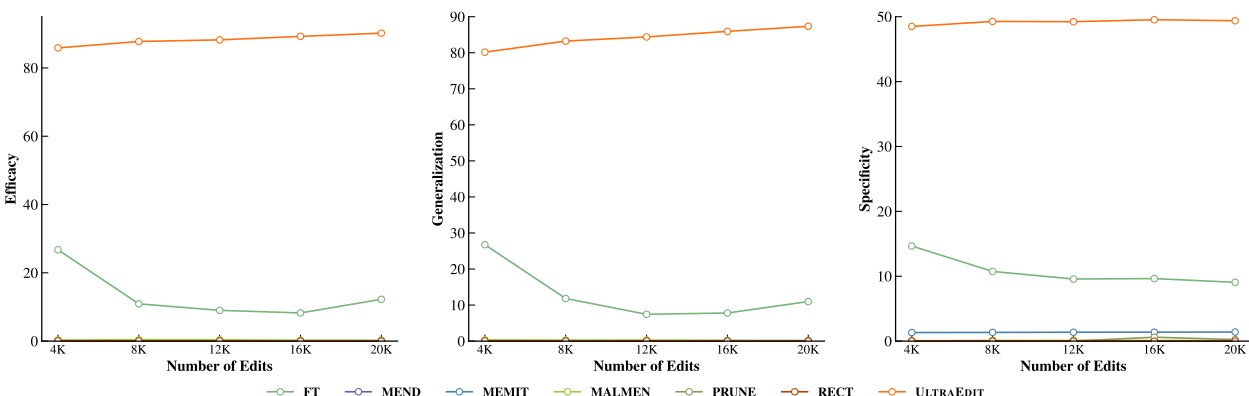

Figure 7: Performance comparison of different baselines as ZsRE edits accumulate in LLaMA-3.

The Generalization and Specificity performance of ULTRAEDIT on larger-parameter models is shown in Figure 8, demonstrating the method's scalability with respect to model size.

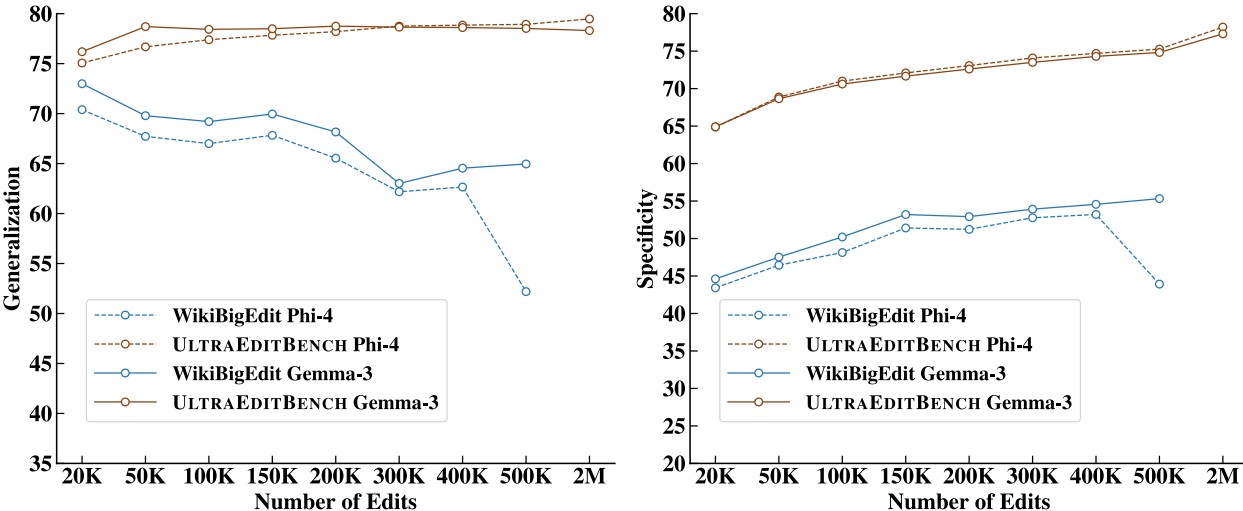

Figure 8: Generalization and Specificity of lifelong editing on Phi-4-14B and Gemma-3-27B.

## C.3   Additional Results on Post-Edited Model Evaluation

We begin by introducing the four evaluation benchmarks used in our study:

**SST** (Stanford Sentiment Treebank) is a sentiment analysis benchmark composed of movie reviews annotated with fine-grained sentiment labels. It evaluates a model's ability to capture subtle emotional cues in natural language.

**MMLU** (Massive Multitask Language Understanding) is a comprehensive benchmark spanning a wide range of academic and professional subjects. It assesses a model's general knowledge and reasoning ability across diverse domains.

**MRPC** (Microsoft Research Paraphrase Corpus) contains sentence pairs annotated for semantic equivalence. This benchmark tests whether a model can accurately identify paraphrases, reflecting its understanding of meaning preservation.

**NLI** (Natural Language Inference) tasks involve determining the logical relationship between a premise and a hypothesis, namely entailment, contradiction, or neutrality. They evaluate a model's capacity for logical reasoning and inference.

Figure 9 provides a full comparison between ULTRAEDIT and baselines, further confirming its minimal effect on the model's inherent abilities.

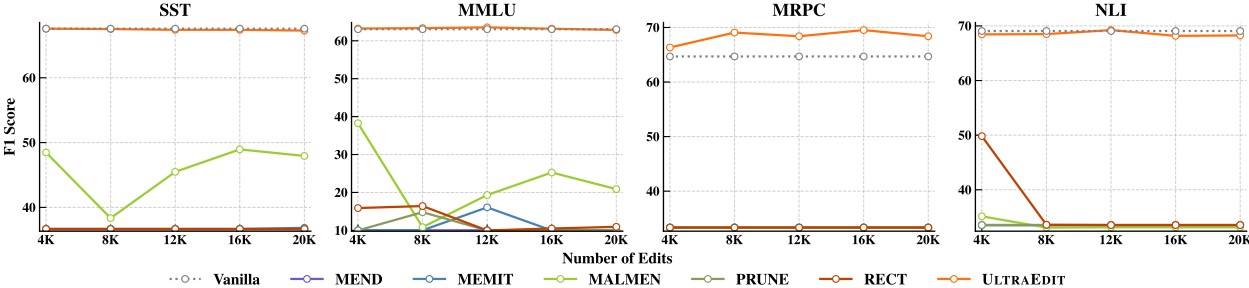

Figure 9: Performance of post-edited LLaMA-3 (20K ZsRE edits) across various benchmark.

## C.4 Failure Scenarios of Gradient-based Localization

Due to structural bias, where gradients disproportionately concentrate on functional tokens (e.g., punctuation or the last token) rather than the semantic subject (often referred to as "attention sinks"); and second, in cases of spurious correlations, where the model relies on non-causal shortcut features (e.g., a specific adjective) to predict the answer, leading gradients to misidentify the editing target. In contrast, explicit masking avoids these distractions by forcibly anchoring updates to the subject, though it requires the restrictive assumption of known subject boundaries.

## D Case study

Table 17: Case study of applying ULTRAEDIT to LLaMA3-8B on ULTRAEDITBENCH.✓ indicates the post-edited output exactly matches the ground-truth answer, while ✗ denotes mismatch.

| Prompt | Pre-edited Output | Ground truth | Post-edited Output |
|---|---|---|---|
| What is the country of citizenship for Olivier Renard? | Oliviergium | belgie | belgie ✓ |
| What type of place is mechanics-ville, delaware? | Mechanics | community | community ✓ |
| What type of ecological system does goobang creek belong to? | Goine | riverine | riverine ✓ |
| What is the profession of shamsula-nuar nasarah? | Shician | Politician | Politician ✓ |
| What type of sport was featured during the 2003 Canadian Open? | The volleyball | indoor tennis | indoor tennis ✓ |
| What is the place of birth of jyotsna radhakrishnan? | Jwaitali | kuwet | malwet ✗ |
| Which location shares a border with guipronvel? | Guzguer | plouguin | Saintouguin ✗ |

