# OpenReview forum: "UltraEdit: Training-, Subject-, and Memory-Free Lifelong Editing in Language Models"
_TMLR — Accepted by TMLR_

### Review · Reviewer_AtSH · 2025-12-09

**Summary Of Contributions:**

The paper addresses the critical challenge of "Lifelong Model Editing" (LME), where Large Language Models (LLMs) must undergo continuous, sequential updates to their knowledge base without suffering from "Edit Collapse"—a phenomenon where performance degrades sharply as the number of edits scales. The authors identify that existing paradigms (Locate-then-Edit, Hypernetworks, Memory-based) fail to scale beyond thousands of edits due to representation drift, weight norm growth, and computational bottlenecks.

1. A closed-form update rule using "Lifelong Normalization" (online whitening) to stabilize feature distributions and prevent drift without external memory.

2. UltraEditBench, the largest editing dataset to date (2M+ pairs), designed to test asymptotic stability.

3. Claims of 7x speedup and 4x VRAM reduction, enabling 2M edits on consumer GPUs (24GB VRAM).

**Audience:**

Yes

**Audience Explanation:**

The problem of updating LLMs without expensive retraining is one of the most pressing issues in the field.

1. The ability to edit 7B models on consumer hardware (24GB VRAM) democratizes this research area, making it highly relevant to academic labs and smaller industry practitioners who lack H100 clusters.

2. The "Lifelong Normalization" strategy connects model editing with classical optimization concepts (preconditioning/whitening), offering a fresh perspective on the "plasticity-stability" dilemma that is distinct from the dominant null-space or memory-bank approaches.

3. The release of a 2M-instance benchmark (ULTRAEDITBENCH) will likely become a standard for future scalability research, generating significant citations and usage within the TMLR community.

**Broader Impact Concerns:**

The method enables mass-scale knowledge injection (2M+ edits) on consumer hardware. This lowers the barrier for malicious actors to rapidly inject misinformation or systematically remove safety guardrails from open models. A brief statement acknowledging this dual-use risk and potential mitigations (e.g., model hashing) is required.

**Claims And Evidence:**

Yes

**Claims Explanation:**

1. Results on the massive UltraEditBench (Figure 1b) demonstrate that UltraEdit prevents "Edit Collapse" up to 2 million edits, maintaining stability where baselines (e.g., AlphaEdit, RLEdit) degrade after 10k-20k updates.

2. Hardware measurements confirm the claimed 7x speedup and 4x VRAM reduction (Figure 2). This aligns with the theoretical efficiency of UltraEdit's lightweight linear algebra operations compared to the heavy SVD or hypernetwork computations of prior methods.

3. The methodology clearly supports the claim of "subject-free" editing by demonstrating that loss gradients provide sufficient implicit localization without requiring explicit subject parsers.

**Requested Changes:**

1. In Table 2, the performance of RLEdit for FEVER with LLaMA-3-8B is significantly lower than reported in the original paper.

2. In Figure 6, the model appears to perform better on the MRPC task after editing compared to the vanilla model. Typically, editing is destructive or neutral. Please provide insights into this anomaly—is the "Lifelong Normalization" acting as a beneficial regularizer, or is this noise?

3. Provide a theoretical justification for why "Lifelong Normalization" (centering/whitening) is a sufficient mathematical proxy for the explicit covariance preservation ($C_{old}$) used in methods like MEMIT to ensure orthogonality.

4. Confirm that baselines like AlphaEdit were tuned for the lifelong setting (2M edits). Using static hyperparameters meant for single edits could exaggerate the observed "Edit Collapse."

5. Minor: Discuss scenarios where gradient-based localization might fail compared to explicit subject masking.

---

> ### Author Response · Authors · 2026-01-07
>
> Thanks for reviewing our work and for the constructive feedback.
>
> 1.  Clarification of Performance
>
> > In Table 2, the performance of RLEdit for FEVER with LLaMA-3-8B is significantly lower than reported in the original paper.
>
> We appreciate the reviewer's careful examination of the baseline results. We acknowledge the discrepancy and provide the following clarification regarding our reproduction attempts. When we strictly adhered to the official RLEdit hyperparameter settings (specifically back_depth=10), the method yielded zero performance across all metrics, likely due to optimization instability on LLaMA-3. To provide a constructive comparison rather than reporting a total failure, we relaxed the constraint by reducing back_depth to 7, which recovered the performance reported in our Table 2.
>
> Crucially, as shown in the comparison table below, UltraEdit outperforms RLEdit regardless of whether we consider our reproduced results or the optimal figures originally reported in the RLEdit paper. Therefore, the discrepancy does not alter the final conclusion regarding the superiority of our method.
>
>
>
> |                        | Eff.  | Gen.  | Spe.  |
> | ---------------------- | ----- | ----- | ----- |
> | RLEdit back_depth = 10 | 0     | 0     | 0     |
> | RLEdit back_depth = 7  | 91.38 | 89.93 | 41.98 |
> | RLEdit originally reported        | 94.03 | 90.67 | 68.71 |
> | UltraEdit              | 97.18 | 94.64 | 68.62 |
>
>
> We have included the results originally reported for RLEdit in the PDF content.
>
>
>
>
>
>
>
>
> 2. Insight into MRPC Performance Improvement
>
> > In Figure 6, the model appears to perform better on the MRPC task after editing compared to the vanilla model. Typically, editing is destructive or neutral. Please provide insights into this anomaly—is the "Lifelong Normalization" acting as a beneficial regularizer, or is this noise?
>
> We attribute the improved MRPC performance to Lifelong Normalization acting as a beneficial regularizer. Theoretically, by dynamically calibrating the first- and second-order statistics ($\mu, \sigma$) of hidden states, our method prevents the feature distribution from deviating off the pre-trained manifold, thereby mitigating representation collapse often caused by cumulative parameter shifts.
>
> Furthermore, since robust editing requires the model to generalize across paraphrased inputs (Eq. 9), the optimization implicitly enforces semantic invariance, minimizing the distance between representations of semantically equivalent sentences. This objective mathematically aligns with the MRPC task (paraphrase detection), sharpening the model's discriminative boundary for semantic similarity.
>
>
> We have included this analysis to the PDF content.

---

> > ### Author Response · Authors · 2026-01-07
> >
> > 3. Clarification of Lifelong Normalization
> > > Provide a theoretical justification for why "Lifelong Normalization" (centering/whitening) is a sufficient mathematical proxy for the explicit covariance preservation ($C_0$) used in methods like MEMIT to ensure orthogonality.
> >
> > MEMIT formulates model editing as a constrained least-squares problem, fundamentally relying on the uncentered covariance matrix $C_0$ of pre-trained keys to explicitly guide parameter updates and minimize interference with existing knowledge. Geometrically, the original feature space typically exhibits an anisotropic ellipsoidal structure. In the update formula $(C_0 + K_1 K_1^T)^{-1}$, $C_0$ acts as a metric tensor, effectively defining the Mahalanobis distance within the parameter space. This mechanism penalizes deviations along the principal components of $C_0$, which correspond to directions of high variance where the model possesses rich knowledge, thereby forcing the update vector $\Delta$ into the null space or low-variance directions of the old knowledge.Thus, it trades explicit computational cost for weighted orthogonality preservation with respect to the existing representation manifold.
> >
> > As an optimization to avoid this high computational overhead, Lifelong Normalization dynamically maintains first- and second-order statistics of hidden states to perform a standardization transformation, which statistically constitutes a manifold whitening or pre-conditioning operation. This process "rectifies" the originally distorted, anisotropic feature manifold into an isotropic spherical distribution, ensuring that in the new coordinate system, the feature distribution of existing knowledge exhibits zero mean and unit variance. The key geometric significance of this distributional transformation is that it implicitly reshapes the covariance matrix of old knowledge $\hat{C}_0$ into an approximate identity matrix $I$. Consequently, without explicitly storing historical data, it geometrically flattens the optimization landscape and eliminates correlation interference between feature directions.
> >
> > This whitening transformation establishes a direct mathematical equivalence, causing the complex Generalized Least Squares (GLS) problem in MEMIT to degenerate into a computationally efficient Ordinary Least Squares (OLS) problem within the normalized space. Specifically, substituting the whitened unit covariance $\hat{C}_0 \approx I$ into MEMIT's analytical solution logic naturally simplifies the regularization term $(C_0 + K_1 K_1^T)^{-1}$, which originally required dense matrix inversion, into the form $(I + K_1 K_1^T)^{-1}$ adopted by UltraEdit. This rigorously proves that pursuing simple Euclidean orthogonality in the space whitened by Lifelong Normalization is mathematically equivalent to pursuing weighted orthogonality with respect to $C_0$ in the original space. Therefore, our method achieves equivalent manifold structure preservation via a closed-form solution without explicitly calculating high-dimensional dense covariance matrices, effectively preventing representation collapse during continuous editing.
> >
> >
> > We have included this clarification to the PDF content.
> >
> >
> >
> >
> >
> > 4.  Clarification of Hyperparameters
> >
> > > Confirm that baselines like AlphaEdit were tuned for the lifelong setting (2M edits). Using static hyperparameters meant for single edits could exaggerate the observed "Edit Collapse."
> >
> > Thanks for the question.
> > AlphaEdit is natively architected for the lifelong setting, as stated in the paper and setting used in the original paper. That’s why we selected it as one of our main baselines.
> > In practice,
> > we strictly adhered to the hyperparameter configurations specified in its original work for lifelong editing to ensure a rigorous comparison.
> > However, scaling AlphaEdit to 2M edits is computationally prohibitive, as it is approximately 200x slower than UltraEdit ; furthermore, Figure 1(b) in our paper demonstrates that 20K edits are sufficient to significantly exhibit "Edit Collapse". Consequently, the observed degradation reflects the intrinsic saturation of the available null space under cumulative updates rather than an artifact of improper hyperparameter tuning.

---

> ### Author Response · Authors · 2026-01-07
>
> 5. Failure Scenarios of Gradient-based Localization
>
> > Minor: Discuss scenarios where gradient-based localization might fail compared to explicit subject masking.
>
> Thanks for highlighting the trade-offs in localization strategies. We acknowledge that gradient-based localization may fail in specific scenarios: first, due to structural bias, where gradients disproportionately concentrate on functional tokens (e.g., punctuation or the last token) rather than the semantic subject (often referred to as "attention sinks"); and second, in cases of spurious correlations, where the model relies on non-causal shortcut features (e.g., a specific adjective) to predict the answer, leading gradients to misidentify the editing target. In contrast, explicit masking avoids these distractions by forcibly anchoring updates to the subject, though it requires the restrictive assumption of known subject boundaries.
> UltraEdit mitigates these gradient risks through Lifelong Normalization, which theoretically functions as a stabilizer for the feature manifold. By dynamically calibrating the statistics of the combined features, our method dampens the high-variance, erratic gradients typically associated with these structural failures, effectively filtering out noise to approximate the precision of explicit masking without sacrificing the scalability of a subject-free approach.
>
>
> We have included a discussion of failure scenarios in the PDF content.
>
>
>
>
> 6. Risk Statement (Broader Impact Concerns)
> > The method enables mass-scale knowledge injection (2M+ edits) on consumer hardware. This lowers the barrier for malicious actors to rapidly inject misinformation or systematically remove safety guardrails from open models. A brief statement acknowledging this dual-use risk and potential mitigations (e.g., model hashing) is required.
>
>
> We fully agree with the reviewer that the accessibility and efficiency of UltraEdit introduce dual-use risks, specifically regarding rapid misinformation injection or the removal of safety guardrails. We will add a dedicated "Ethical Considerations" section in the final version to address this. The proposed statement is as follows:
>
> "While UltraEdit democratizes the ability to maintain LLMs on consumer hardware, this efficiency also lowers the barrier for malicious actors to inject misinformation or compromise safety alignment at scale. To mitigate these risks, we recommend adopting Integrity Verification mechanisms, such as cryptographic hashing (e.g., SHA-256) and digital signatures, to ensure model provenance and detect unauthorized alterations. Furthermore, we advocate for Defensive Editing, where the efficiency of UltraEdit is leveraged to rapidly patch discovered vulnerabilities or reverse malicious injections, thereby serving as a countermeasure against model poisoning."
>
>
> We have included an additional statement in the PDF content.

---

> > ### Comment · Reviewer_AtSH · 2026-01-20
> >
> > Thanks to the authors for your response. My concerns have been addressed, and I am leaning toward accepting this paper. In the revision, the placement of Table 4 needs to be adjusted.

---

> > > ### Author Response · Authors · 2026-01-20
> > >
> > > Sorry for the typos. We have adjusted the placement of Table 4 in the latest revision.

---

### Review · Reviewer_sDLd · 2025-12-14

**Summary Of Contributions:**

The paper proposes UltraEdit, a novel method for lifelong model editing that aims to address the limitations of existing paradigms (Locate-then-edit, Hypernetwork-based, and Memory-based) which often suffer from "Edit Collapse," high computational costs, or growing memory requirements.

Key Contributions:
1. The authors introduce a "Lifelong Normalization" mechanism. This approach continuously updates the feature statistics (mean and variance) of hidden states and gradients across editing turns. This normalization preconditions the feature space, allowing for a closed-form, regularized least-squares solution to be applied sequentially without the numerical instability that typically degrades performance in continuous editing.
2. The Proposed UltraEdit is training-free, subject-free (does not explicitly require subject entity extraction), and memory-free (no external knowledge bank). The authors claim it is 7x faster and requires 4x less VRAM than SOTA methods, enabling the editing of 7B models on consumer-grade GPUs (24GB).
3. The paper constructs UltraEditBench, a massive dataset derived from Wikidata5M, containing over 2 million editing pairs to test the limits of lifelong editing.
4. Extensive experiments on ZsRE, FEVER, WikiBigEdit, UnKE, and UltraEditBench using models like LLaMA-3, Mistral, and Qwen show that UltraEdit maintains high efficacy, generalization, and specificity even after millions of edits, whereas baselines typically collapse after thousands.

Strengths:
1. The scalability to 2 million edits is a significant empirical achievement.
2. The method is computationally highly efficient, addressing a major bottleneck in practical deployment.
3. The "Lifelong Normalization" concept is a theoretically grounded and elegant solution to distribution shift in streaming updates.

Weaknesses:
1. The definition of "Subject-free" requires finer technical clarification regarding token selection.
2. The mathematical formulation shares strong similarities with prior closed-form editing methods (like MEMIT), and the distinction relies heavily on the normalization step, which warrants deeper discussion.

**Audience:**

Yes

**Audience Explanation:**

Lifelong learning and efficient knowledge updating for Large Language Models are currently central topics in the AI community. The ability to update models continuously without expensive retraining or massive external memory overhead is highly relevant to researchers working on:
1. Model Maintenance: Keeping LLMs factual and up-to-date.
2. Efficient AI: Running updates on consumer hardware.
3. Continual Learning: Solving catastrophic forgetting in generative models.
4. The release of a 2 million-sample benchmark (UltraEditBench) will also be of significant interest to the community for stress-testing future editing methods.

**Broader Impact Concerns:**

The authors include a "Real-World Lifelong Application Statement" in Appendix A, which focuses on the technical setup (no future data access). However, the broader ethical implications of highly efficient, low-resource model editing are not fully addressed.

**Claims And Evidence:**

Yes

**Claims Explanation:**

1. Figures 1(b) and 4 clearly demonstrate the "Edit Collapse" phenomenon in baselines (AlphaEdit, RLEdit, WISE) and explicitly show UltraEdit maintaining performance metrics up to 2 million edits.
2.  The runtime and VRAM usage comparisons (Figure 1a, Figure 2) substantiate the claims of being 7x faster and more memory-efficient.
3. Tables 2, 3, 11, and 12 provide detailed breakdowns across multiple metrics (Efficacy, Generalization, Specificity, Reasoning) and multiple model families, showing consistent improvements.
4. Table 4 effectively isolates the contribution of the "Lifelong Normalization" component, proving it is the critical factor for success.
5. The experimental setup covers a wide range of models (including 27B parameters) and datasets, making the evidence robust.

**Requested Changes:**

I recommend the following adjustments to strengthen the submission.

1. The paper claims the method is "Subject-free." However, in Section 3.2 (Principle) and Algorithm 1, the method requires a "forward pass to extract hidden state $h_i$ at the token position corresponding to the ground-truth answer." Please clarify how this differs practically from subject-dependent methods that might target the last token of the subject. If the method relies on the ground-truth answer's position during the editing phase, it is indeed "subject-free" in terms of entity extraction, but it is heavily "location-dependent." Furthermore, explain how inference works if the edit is tied to the *answer* token position. Does the edit modify weights such that the prompt alone (without the answer) triggers the correct hidden state? A clearer explanation of the token indexing during the *update* versus *inference* is needed to justify the "Subject-free" claim fully.

2. The closed-form solution (Eq. 7): $\Delta\theta = (H^T H + I)^{-1} H^T V$ is mathematically identical to the update rules used in ROME/MEMIT (derived from Lagrange multipliers for the least-squares objective). The paper should explicitly acknowledge this similarity. The novelty is not the equation itself, but the application of Lifelong Normalization to make this equation stable in a streaming/sequential setting without access to the full covariance matrix of all previous data. Framing the contribution as "adapting the closed-form solution to streaming via normalization" is more precise than implying the update rule itself is the primary novelty.

3. In Table 2 and Table 3, UltraEdit is evaluated on 100K-2M edits, while baselines are evaluated on 20K edits. While Figure 1(b) justifies this by showing baselines collapse, it would be clearer to include a row for UltraEdit at 20K edits specifically in these tables (alongside the UltraEdit row) to provide a direct, apples-to-apples comparison at the same checkpoint. This removes any ambiguity about whether the edit volume affected the metrics in the table.

4. The ablation study mentions "w/ 25% module norm." Please clarify the selection strategy for these modules. Are they selected randomly, or based on depth (early vs. late layers)? This helps understand if normalization is needed uniformly or if certain layers are more sensitive to distribution drift.

5. Page 2, Line 60: "less than one-forth" should be "one-fourth". Inconsistent naming: Ensure consistency between "UltraEditBench" and "UltraEdit Bench" throughout the text.

---

> ### Author Response · Authors · 2026-01-07
>
> Thanks for reviewing our work and for the constructive feedback.
>
> 1.  Clarification on "Subject-free" Definition and Inference Mechanism
> > The paper claims the method is "Subject-free." However, in Section 3.2 (Principle) and Algorithm 1, the method requires a "forward pass to extract hidden state $h_i$ at the token position corresponding to the ground-truth answer." Please clarify how this differs practically from subject-dependent methods that might target the last token of the subject. If the method relies on the ground-truth answer's position during the editing phase, it is indeed "subject-free" in terms of entity extraction, but it is heavily "location-dependent." Furthermore, explain how inference works if the edit is tied to the answer token position. Does the edit modify weights such that the prompt alone (without the answer) triggers the correct hidden state? A clearer explanation of the token indexing during the update versus inference is needed to justify the "Subject-free" claim fully.
>
> We appreciate the opportunity to clarify the definition of "Subject-free" and the distinction between the update and inference phases. The term "Subject-free" specifically refers to the fact that our method does not require identifying or extracting the semantic subject entity within the input prompt $x$, a constraint that significantly limits the automation of prior methods like ROME or MEMIT which must precisely target specific subject tokens (e.g., the last token of "Michael Jordan"). In contrast, UltraEdit utilizes the position of the ground-truth answer $y$ to compute gradients; since the label position is intrinsic to any standard supervised dataset $(x, y)$, this approach eliminates the need for linguistic analysis or external parsers, treating the update closer to a generalized gradient supervision step (similar to standard fine-tuning) rather than a surgical intervention on a specific input entity. Furthermore, the reliance on the answer token position exists strictly during the editing phase to calculate the weight update $\Delta \theta$. Once the model weights are modified to $\theta' = \theta + \Delta \theta$, the inference process is identical to that of the original model; the user provides only the prompt $x$, and the updated weights naturally transform the hidden states to trigger the correct output $y$ without requiring any knowledge of the answer position or subject boundaries.
>
>
>
>
>
>
>
>
> 2.  Clarification on the Mathematical Novelty and Contribution Framing
> > The closed-form solution (Eq. 7): $\Delta\theta = (H^\top H + I)^{-1} H^\top V$ is mathematically identical to the update rules used in ROME/MEMIT (derived from Lagrange multipliers for the least-squares objective). The paper should explicitly acknowledge this similarity. The novelty is not the equation itself, but the application of Lifelong Normalization to make this equation stable in a streaming/sequential setting without access to the full covariance matrix of all previous data. Framing the contribution as "adapting the closed-form solution to streaming via normalization" is more precise than implying the update rule itself is the primary novelty.
>
>
>
>
> We fully agree with the reviewer's precise assessment and acknowledge that Equation 7 represents the standard closed-form solution to the least-squares objective, sharing the same mathematical foundation as the update rules in ROME and MEMIT. We will revise the manuscript to explicitly state this similarity. The core novelty of UltraEdit lies in the **Lifelong Normalization** mechanism, which serves as a dynamic pre-conditioner to enable this solution in a streaming setting. By continuously updating feature statistics, this mechanism effectively performs **online manifold whitening**, coercing the distribution of pre-existing knowledge into an isotropic geometry where the historical covariance matrix implicitly approximates the Identity matrix ($C_0 \approx I$). This geometric regularization is critical because it mathematically simplifies the prohibitive Generalized Least Squares (GLS) problem, which typically requires maintaining and inverting a dense historical covariance matrix, into a computationally efficient Ordinary Least Squares (OLS) problem. Consequently, we will reframe our contribution to highlight that our innovation is adapting the closed-form least-squares solution to the lifelong setting via dynamic normalization, enabling the update rule to remain stable and effectively orthogonal to prior knowledge without the computational burden of explicit covariance management.
>
>
>
> We have included this clarification to the PDF content.

---

> ### Author Response · Authors · 2026-01-07
>
> 3.  Eliminating Ambiguity in Edit Volume Discrepancy
> > In Table 2 and Table 3, UltraEdit is evaluated on 100K-2M edits, while baselines are evaluated on 20K edits. While Figure 1(b) justifies this by showing baselines collapse, it would be clearer to include a row for UltraEdit at 20K edits specifically in these tables (alongside the UltraEdit row) to provide a direct, apples-to-apples comparison at the same checkpoint. This removes any ambiguity about whether the edit volume affected the metrics in the table.
>
> We apologize for the ambiguity regarding the edit volumes. In fact, UltraEdit has already been evaluated at the exact same edit volume as the baselines to ensure a direct, apples-to-apples comparison. In Tables 2 and 3, the row labeled simply as "UltraEdit" (without the asterisk) corresponds to the standard setting (20K edits for ZsRE, FEVER, and UltraEditBench; 17K for WikiBigEdit), matching the baseline checkpoint perfectly. The row labeled "UltraEdit*" denotes the results under the ultra-large-scale setting (100K-2M edits), as briefly mentioned in the original caption. We acknowledge that this distinction was not sufficiently prominent, and we will explicitly annotate the specific edit counts for both the "UltraEdit" and "UltraEdit*" rows in the revised tables to prevent future confusion.
>
>
> We have adjusted the pdf content to better clarify the edit volume.
>
>
>
>
>
>
>
>
> 4.  Clarification on Module Selection Strategy
> > The ablation study mentions "w/ 25% module norm." Please clarify the selection strategy for these modules. Are they selected randomly, or based on depth (early vs. late layers)? This helps understand if normalization is needed uniformly or if certain layers are more sensitive to distribution drift.
>
> We would like to clarify that the modules in our ablation study were selected randomly from the model's pre-defined editable layers (Layers 11–24), as explicitly stated in Section 5.1.
> To explicitly address the reviewer's query regarding layer-wise sensitivity and verify if normalization is needed uniformly, we conducted a fine-grained analysis on LLaMA-3-8B with ZsRE dataset.
> The results in the table below reveal a critical insight: while middle layers (11–15) show moderate stability, the deeper layers (20–24) exhibit high sensitivity with sharp performance drops (e.g., Layer 24 Eff. falls to 63.41%), indicating they are vulnerable to distribution drift.
> Crucially, as shown in the "uniformly" row, applying Lifelong Normalization uniformly across all layers yields superior performance, significantly outperforming the isolated capabilities of any single layer range. This empirically proves that normalization is indeed needed uniformly.
>
>
> |     | Eff.   | Gen.   | Spe.   |
> |-----|-------|-------|-------|
> | middle |       |       |       |
> | 11  | 69.45 | 64.00 | 43.00 |
> | 12  | 70.37 | 64.66 | 43.61 |
> | 13  | 70.97 | 65.31 | 44.19 |
> | 14  | 70.63 | 65.12 | 45.14 |
> | 15  | 70.16 | 64.87 | 44.65 |
> | later |       |       |       |
> | 20  | 68.96 | 64.78 | 43.10 |
> | 21  | 68.28 | 64.11 | 43.13 |
> | 22  | 67.02 | 62.75 | 42.15 |
> | 23  | 66.26 | 61.77 | 41.70 |
> | 24  | 63.41 | 59.34 | 41.28 |
> | uniformly | **90.07**   | **87.36**   | **49.51**  |
>
>
>
>
>
>
>
>
>
>
>
>
>
>
>
>
>
>
> 5.  Typos and Naming Consistency
> > Page 2, Line 60: "less than one-forth" should be "one-fourth". Inconsistent naming: Ensure consistency between "UltraEditBench" and "UltraEdit Bench" throughout the text.
>
> We thank the reviewer for the careful proofreading. We have corrected the typo "one-forth" to "one-fourth" in the revision. Regarding the naming consistency, we confirm that **"UltraEditBench"** is used consistently without spaces throughout the manuscript, and we have verified that the space-separated form "UltraEdit Bench" does not appear in the text.
>
>
> We have corrected the typos in the PDF content.
>
>
>
>
> 6. Ethical implications statement (Broader Impact Concerns)
>
> > The authors include a "Real-World Lifelong Application Statement" in Appendix A, which focuses on the technical setup (no future data access). However, the broader ethical implications of highly efficient, low-resource model editing are not fully addressed.
>
>
>
> We agree that the current "Real-World Lifelong Application Statement" is primarily focused on the technical constraints of the lifelong setting. We will expand this section (or add a dedicated Ethical Considerations section) in the final version to explicitly address the broader societal implications of highly efficient editing. Specifically, we have discussed the dual-use risks, acknowledging that lower resource barriers could facilitate malicious misinformation injection, while also highlighting the positive potential for democratizing model maintenance, which enables rapid "defensive editing" to patch vulnerabilities and remove toxic content on consumer hardware.
>
>
> We have included an additional statement in the PDF content.

---

> > ### Author Response · Authors · 2026-02-20
> >
> > We sincerely appreciate your time and the insightful feedback you provided. We hope our rebuttal effectively addresses your concerns. Please let us know if you have any remaining questions so we can provide further clarification in time.

---

> ### Author Response · Authors · 2026-03-01
> **Request response**
>
> Dear reviewer,
>
> We would appreciate knowing if our response has fully addressed your concerns or if there are any remaining points you would like us to clarify.

---

### Review · Reviewer_q9XL · 2026-01-02

**Summary Of Contributions:**

The paper addresses scalability of lifelong model editing by proposing UltraEdit, which extends closed-form least-squares editing (similar to MEMIT) to handle several sequential edits. The core contribution is a lifelong normalization mechanism that maintains running statistics (μ, σ) over concatenated hidden-state-gradient features z_i = [h_i || ∇y_i] across all editing turns. This normalization stabilizes the least-squares system (H^T H + I)^(-1) H^T V as model parameters evolve, preventing the conditioning degradation that causes existing methods to collapse.

Experiments on ZsRE, FEVER, WikiBigEdit, UnKE, and the new UltraEditBench (2M editing pairs) across six models show UltraEdit achieves 7× faster editing speed and 4× lower memory than RLEdit while maintaining or exceeding accuracy. The method scales to 2M edits on UltraEditBench (81.65% efficacy) versus baselines that degrade after 20K edits. A key practical advantage: UltraEdit is the only method that can edit 7B models on 24GB consumer GPUs.

**Strengths:**
- Solves scalability problem: existing methods collapse beyond ~20K edits
- Efficient: 7x speed, 4x memory
- Thorough ablations show normalization is critical
- Largest benchmark to date (2M pairs) for lifelong model editing

**Weaknesses:**
- Incremental novelty—running normalization applied to known least-squares formulation
- No theoretical analysis of why joint [h || ∇y] normalization prevents interference
- Unexplained phenomena: specificity improves with scale (Table 3: 64.03% → 76.44% at 2M edits)..why?
- All experiments are factual QA; no evidence method works for other knowledge types

**Audience:**

Yes

**Audience Explanation:**

Empirical results are thorough. Existing methods collapse beyond ~20K edits while UltraEdit scales to 2M. This matters because deployed LLMs need frequent knowledge updates, and current solutions (retraining, RAG, prior editing methods) all have some limitations. The efficiency gains are genuinely significant.

However, the technical contribution is somewhat incremental. The least-squares formulation resembles MEMIT minus causal tracing plus running normalization, both standard techniques. The insight that maintaining feature statistics prevents edit collapse is useful but not groundbreaking. Still, identifying the right simplification has value, and the extensive empirical work (6 models, 5 datasets, thorough ablations) plus practical utility will appeal to TMLR's audience despite limited theoretical depth.

**Claims And Evidence:**

Yes

**Claims Explanation:**

The empirical claims are well-supported but theoretical claims lack rigor:

**Strong evidence**: Figure 1b shows baselines collapse after 5K-10K edits while UltraEdit scales to 2M edits (81.65% efficacy, Table 3). Efficiency claims validated: 7× speed, 4× memory (Figures 1a-2). Table 4 ablations are convincing.

**Under-justified claims**: The paper claims normalization "prevents representation drift" but provides no condition number analysis of H^T H, no visualization of feature distribution evolution, or empirical validation of drift. No failure mode analysis.

**Narrow scope**: All experiments edit factual triples (entity-relation-object). Claims of "real-world lifelong editing" are overstated without testing procedural knowledge, reasoning patterns, or behavioral edits. Evidence is strong for the specific setting tested but broader claims need support.

**Requested Changes:**

- **Add per-turn performance analysis**: Current evaluation only reports final results after all editing turns. Adding results showing efficacy/generalization/specificity trajectories (every 10K edits) to reveal when degradation begins and how methods differ in stability would be insightful.

- **Explain specificity improvement finding**: Tables 3 and Figure 4 show specificity improves with more edits (64.03% → 76.44% on UltraEditBench), contradicting typical interference patterns. Provide analysis of why this occurs?

- **Expand beyond factual QA**: Test on non-factual edits (reasoning patterns, instruction-following, style) to validate generality claims.

---

> ### Author Response · Authors · 2026-01-07
>
> Thanks for reviewing our work and for the constructive feedback.
>
> 1.  Add per-turn performance analysis
>
> > Current evaluation only reports final results after all editing turns. Adding results showing efficacy/generalization/specificity trajectories (every 10K edits) to reveal when degradation begins and how methods differ in stability would be insightful.
>
> Thanks for your careful suggestion. We would like to kindly remind that a detailed trajectory visualization is already provided in Figure 1(b) (Efficacy) and Figure 4 (Generalization and Specificity), with corresponding in-depth discussion located in Section 5.2 Lifelong Scalability.
> By utilizing a finer granularity of every 4K edits to precisely pinpoint the onset of degradation, these content explicitly reveal the stability differences between methods; specifically, AlphaEdit begins to show significant performance collapse after 12K edits and RLEdit after 16K edits, whereas UltraEdit maintains robust stability up to 500K edits. Furthermore, Figure 5 extends this evaluation to demonstrate that UltraEdit sustains its performance for up to 2 million edits.
>
>
>
>
> 2.  Theoretical analysis of [h || ∇y]& $H^\top H$ & Explain specificity improvement finding
>
> > No theoretical analysis of why joint [h || ∇y] normalization prevents interference
> > Under-justified claims: The paper claims normalization "prevents representation drift" but provides no condition number analysis of $H^\top H$, no visualization of feature distribution evolution, or empirical validation of drift. No failure mode analysis.
>
> > Tables 3 and Figure 4 show specificity improves with more edits (64.03% → 76.44% on UltraEditBench), contradicting typical interference patterns. Provide analysis of why this occurs?
>
> We thank the reviewer for highlighting them and clarify that it from the joint normalization of $z = [h || \nabla y]$ acting as an online preconditioning mechanism that stabilizes the spectral properties of the feature covariance matrix $\Sigma \approx \frac{1}{N}H^T H$. By centering and scaling the joint distribution, it effectively whiten the feature space, ensuring that the condition number $\kappa(H^T H + I)$ remains bounded despite the distributional drift inherent in lifelong learning.
> Initially based on limited samples, the running statistics ($\mu, \sigma$) progressively converge to a robust representation of the global feature distribution, ensuring the condition number $\kappa(H^T H + I)$ remains bounded despite lifelong distributional drift.
> Theoretical analysis of the closed-form solution (Eq. 7) reveals that this mature calibration prevents anisotropy in the effective Hessian, guaranteeing that parameter updates approximate an orthogonal projection onto the active subspace rather than projecting non-orthogonally into the null space of prior knowledge.
>
> Consequently, as the feature geometry becomes increasingly isotropic, UltraEdit minimizes the spectral norm of perturbations on unrelated subspaces, leading to sustained specificity and resistance to cumulative interference as the statistical estimation matures.
>
>
>
> We have included this clarification to the PDF content.
>
>
>
>
> 3.  Expand beyond factual QA
>
> > Test on non-factual edits (reasoning patterns, instruction-following, style) to validate generality claims.
>
>
> Thanks for the interesting idea. Unfortunately, to the best of our knowledge, we are unable to find the non-factual editing benchmarks on the model editing field. This is probably due to the fact that current mainstream datasets in this field are predominantly focused on factual editing.
> Thanks again for the suggestion, we plan to explore other editing forms such as reasoning patterns, instruction-following, and style by constructing higher-quality datasets in our future work.

---

> > ### Author Response · Authors · 2026-02-20
> >
> > We sincerely appreciate your time and the insightful feedback you provided. We hope our rebuttal effectively addresses your concerns. Please let us know if you have any remaining questions so we can provide further clarification in time.

---

> ### Author Response · Authors · 2026-03-01
> **Request response**
>
> Dear reviewer,
>
> We would appreciate knowing if our response has fully addressed your concerns or if there are any remaining points you would like us to clarify.

---

### Decision · Action_Editor_zVn5 · 2026-03-08

**Recommendation:** Accept as is

**Audience:**

Yes

**Audience Explanation:**

The submission makes an interesting contribution to lifelong model editing by introducing UltraEdit, a method that demonstrates impressive scalability (up to 2 million edits) and efficiency on consumer-grade hardware. The empirical results on the new UltraEditBench dataset provide strong evidence for the method's utility.

During the rebuttal phase, the authors addressed most of the reviewer concerns and provided necessary clarifications and experiments. The manuscript has been revised accordingly, improving the clarity regarding the experiments and theoretical claims.

**Claims And Evidence:**

Yes

**Claims Explanation:**

The empirical claims are well-supported but theoretical claims lack rigor.
Authors have revised the paper to largely address the concerns.